



# Impacts of $H_2O$ variability on accuracy of $CH_4$ observations from MIPAS satellite over tropics

Temesgen Yirdaw Berhe[1], Gizaw Mengistu Tsidu[1,2], Thomas Blumenstock[3], Frank Hase[3], Thomas von Clarmann[3], Justus Notholt[4], and Emmanuel Mahieu[5]

[1]Department of Physics, Addis Ababa University, P.O. Box 1176, Addis Ababa, Ethiopia

[2] Department of Earth and Environmental Sciences, College of Science, Botswana International University of Technology and Science (BIUST), Priv.Bag 16, Palapye, Botswana

[3]Institute of Meteorology and Climate Researchs (IMK-ASF), Karlsruhe Institute of Technology (KIT), Karlsruhe, Germany.

[4]Institute of Environmental Physics, University of Bremen, Bremen, Germany.

[5]Institut d'Astrophysique et de Géophysique, University of Liège (ULg), Liège, Belgium.

**Correspondence:** T. Yirdaw Berhe
(temiephys@gmail.com)

**Abstract.** Uncertainties of tropical methane concentrations, retrieved from spectra recorded by the Michelson Interferometer for Passive Atmospheric Sounding (MIPAS), MIPAS version V5R_CH4_220 are large. We explore the relation of these uncertainties with water vapour variability. We further show that these uncertainties have been reduced in MIPAS version V5R_CH4_224. Coincident measurements of $CH_4$ by MIPAS, ground based FTIR and $CH_4$ derived from EOS MLS coinci-

dent measurements of atmospheric water vapour ($H_2O$), carbon monoxide (CO) and nitrous oxide ($N_2O$) are used to estimate the standard uncertainty of MIPAS CH4 220, MIPAS CH4 224 and natural variability of $H_2O$. Different methods such as bias evaluation, differential method and correlation coefficient are employed to explore the latitudinal variations of standard uncertainty of MIPAS CH4 220 and natural variability of water vapour as well as its reduction on MIPAS CH4 224. The averaged bias between MIPAS CH4 220 and ground-based FTIR measurements are -12.3 %, 8.4 % and 1.2 % for tropics, mid-latitudes

and high latitudes, respectively. The standard deviations of the differences for these latitudinal bands are 5.9 %, 4.8 %and 4.7 %. Moreover, the correlation coefficient between MIPAS CH4 220 and MIPAS V5R_N2O_220 is 0.32 in the upper troposphere and lower stratosphere over tropics and larger than the modest value 0.5 in mid and high latitudes. The poor correlation between MIPAS CH4 220 and MIPAS N2O 220 over tropics can indicate the large uncertainty of MIPAS CH4 220 over tropics that is related to water variability. Similarly, mean relative difference between MIPAS CH4 224 and ground-based FTIR measure-

ments are 3.9 %, -2.6 % and -2.7 % in altitude 15-21 km and the average estimated uncertainty of MIPAS CH4 224 methane were obtained 2.4 %, 1.4 % and 5.1 % in altitude ranges of 15 to 27 km for tropics, mid and high latitudes, respectively. The estimated measurement uncertainty of MIPAS CH4 224 is different for the three latitude bands in the northern hemisphere, reflecting the latitudinal variation of uncertainties of MIPAS methane. However, the large reduction of uncertainty in MIPAS CH4 224 as compared to MIPAS CH4 220 has been confirmed for the tropical measurements. The correlation coefficients

between the uncertainty of MIPAS CH4 220 and the variability of water vapour in lower stratosphere are strong (0.88) on





monthly temporal scales. Similar methods were used for MIPAS CH4 224. It was found that the uncertainty in methane due to the variability of water vapor has been reduced.

## 1 Introduction

Water vapour is a strong absorber in the infrared with high spatiotemporal variability as compared to other greenhouse gases
(Vogelmann et al., 2015). However, much of the stratospheric variability of water vapour in the tropics is a result of oxidation of methane (Nassar et al., 2005). However, the impacts of $H_2O$ variability on the measurements of other greenhouse gases have not yet been studied. We investigate the influence of water vapor variability of methane retrievals from spectra recorded with the Michelson Interferometer for Passive Atmospheric Sounding (MIPAS, Fischer et al. 2008) performed at the institute of Meteorology and Climate Research (IMK) in cooperation with the instito de Astrofía (IAA). We use MIPAS data version
V5R_CH4_220 (von Clarmann et al., 2009) and compare these with the more recent data version V5R_CH4_224 (Plieninger et al., 2016). Although uncertainties in the latter data product have been largely reduced, measurements of methane at tropical sites are still difficult. This is because in the tropics the upper troposphere and lower stratosphere (UT/LS) is very humid and strong vertical gradients are prevailing (Schneider et al., 2006). Evidence of substantial diurnal variations of atmospheric water vapour has been exhibited as reported in Wang et al. (2007) and the spatiotemporal variability of water vapour on the upper
troposphere has already been examined in detail (WMO , 2000; Vogelmann et al., 2015).

Methane and nitrous oxide are produced at the surface and they are not directly coupled chemically. In the UT/LS, the mixing ratios of these long-lived trace gases are largely controlled by dynamical processes, generally resulting in compact tracer-tracer correlations. These correlations are usually more compact in high and mid-latitudes, while in tropics a somewhat larger scatter is observed (Plumb et al., 2007; Payan et al., 2009). In this paper we inquire into the bias and random error of the MIPAS
methane products.

In the lower stratosphere (below 25 km) MIPAS methane has long been known to be biased high (see e.g., Laeng et al. 2015). However, The retrieval setup has been improve, leading to a smaller bias in MIPAS CH4 224 (Plieninger et al., 2016). In the new set up, $H_2O$ concentrations are included in the list of species fitted along with the target species, reducing the propagation $H_2O$ a priori assumptions onto the $CH_4$ profile. A high bias and large random uncertainties in the tropical UT/LS
are not a feature of the IMK methane only. Also in the operational ESA data product, a high bias in MIPAS methane has also been reported for the operational MIPAS data product of ESA (Payan et al., 2009; Errera et al., 2016). Moreover, high random uncertainty of MIPAS ESA $CH_4$ product in the tropical upper troposphere and lower stratosphere atmospheric conditions than the mid and high latitude condition has also been revealed in (Payan et al., 2009; Errera et al., 2016). On the other hand, a large variability of water vapour in the upper troposphere and lower stratosphere in the tropics was reported by Schneider et
al. (2006) and William et al. (2013). Moreover, there are also some studies that hypothesize that the large uncertainty of the MIPAS tropical $CH_4$ product by ESA is caused by the large variability of water vapour even though they did not quantified the contributions (e.g. Payan et al. 2009).





Therefore, this study aims to assess the latitudinal variation of MIPAS CH4 220 and MIPAS CH4 224 uncertainty. Furthermore, we analyze the relationship between these uncertainties and the variability of water vapor. The coincident measurements of $H_2O$, $CH_4$ and $N_2O$ by MIPAS, ground based FTIR and $CH_4$ derived from EOS MLS coincident measurements of atmospheric water vapour ($H_2O$), carbon monoxide (CO) and nitrous oxide ($N_2O$) were used to estimate the uncertainty of MIPAS CH4 220 and MIPAS CH4 224 vertical profiles or integrated columns and the natural variability of $H_2O$ over tropics.

The paper is organized in five sections. The following section includes the description of the data sets used. In sections 3 our analysis method is presented. The results are presented in section 4. In section 5 we summarize our findings.

## 2    Data and instruments

### 2.1    MIPAS Data Sets

The Michelson Interferometer for Passive Atmospheric Sounding (MIPAS) instrument is a high-resolution atmospheric limb

sounder aboard ESA's ENVISAT launched in March 2002 and operating in a sun-synchronous orbit, which delivered limb spectra of atmospheric infrared of a range 685 cm$^{-1}$ to 2410 cm$^{-1}$ along with a resolution of of 0.035 cm$^{-1}$. It aims at global and simultaneous measurements of the chemical composition of the middle atmosphere and upper troposphere. The pointing system allows MIPAS to observe atmospheric parameters in a maximum altitude range of 5-160 km with a vertical spacing of 1-8 km depending on the altitude and the measurement mode. All species are retrieved on a fixed altitude grid, using a

grid width of 1 km from 0 to 44 km, 2 km from 44 to 70 km, 5 km from 70 to 80 km. The infrared limb spectra are inverted to provide profiles of numerous trace gases, including $CH_4$ and $N_2O$ (Fischer et al., 2008). In this study, we have used the reduced spectral resolution (Institute of Meteorology and Climate Research) IMK/IAA MIPAS data product V5R_CH4_224, V5R_N2O_224 (Plieninger et al., 2015) and V5R_CH4_220, V5R_N2O_220 and V5R_H2O_220 (von Clarmann et al., 2009) to achieve the objective of this paper.

### 2.2    Microwave Limb Sounder (MLS) Data Sets

The Earth Observing System (EOS) Microwave Limb Sounder (MLS) is one of four instruments on the NASA's EOS Aura satellite, launched on July 15, 2004 into a near polar sun-synchronous orbit at 705 km altitude (Schoeberl et al., 2006). The spatial coverage of this instrument is nearly global (-82°S to 82° N) and individual profiles are spaced horizontally by 1.5$^o$ or 165 km along the orbit track. In this work, we have used version 3.3 MLS of $N_2O$ and $H_2O$ volume mixing ratio Lambert et

al. (2007). Beyond this, we have used a $CH_4$ data product which was generated from MLS $H_2O$, CO and $N_2O$ measurements Minschwaner et al. (2015). The vertical resolution of MLS $CH_4$ is between 4 and 5 km and 4 to 6 km for $N_2O$. $H_2O$ is retrieved from measurements of the 183 GHz $H_2O$ rotational line spectrum (Lambert et al., 2007; Read et al., 2007). Selection criteria were implemented as suggested by Livesey et al. (2013). The valid pressure range of 100 to 0.46 hPa and non flagged data were considered. More details regarding the MLS experiment and data screening are provided in the above references in detail

and at http://mls.jpl.nasa.gov/data/datadocs.php.



## 2.3 Ground based Fourier transform infrared (FTIR) Data

Network for the Detection of Atmospheric Composition Change (NDACC) is a global network community that monitors changes in atmospheric composition. Fourier transform infrared (FTIR) spectrometers are operated at various stations worldwide on a regular basis and provides long-term observations of many trace gases to asses their impact on global climate. In this

work, we will present results from data recorded at Addis Ababa, Ethiopia (9.01° N, 38.77° E, 2443 m a.s.l.), Jungfraujoch, Switzerland (46.5° N, 8.0° E, 3580 m a.s.l.) and NyÅlesund, Spitsbergen (78.92° N, 11.9° E, 20 m a.s.l.) as stations located in three different latitude bands. The time periods under considerations were starting May, 2009 to Feb, 2011 for Addis Ababa, Jan., 2009 to Feb., 2011 for Jungfraujoch and March, 2009 to April, 2011 for NyÅlesund are considered in this paper. The profiles were retrieved using the inversion code PROFFIT (PROFile FIT) (Hase et al., 2004) for Addis Ababa, but the retrievals

of the remaining two stations have been performed using the SFIT2 algorithm. Both PROFFIT and SFIT2 codes have been mutually validated successfully (Hase et al., 2004).

## 3 Methodology

For bias and precision validation of MIPAS CH4 we use established methodology. The bias is calculated as the mean difference between the data sets, and its significance is assessed via the standard deviation of the mean difference of collocated measure-

ments (von Clarmann, 2006). The precision is estimated as the standard deviation of the differences between the collocated measurements (Stiller et al., 2012). Furthermore, we use the method proposed by Fioletov et al. (2006) which estimates both the natural variability of a state variable and the random uncertainties of two systems measuring this state variable in the same latitude band.

In the case of bias evaluation, the satellite measurement profiles are smoothed using the FTIR averaging kernels of individual

species obtained from the ground based FTIR retrieval by applying the procedures reported in Rodgers and Connor (2003) and given as

$$\mathbf{x}_s = \mathbf{x}_a + \mathbf{A}(\mathbf{x}_i - \mathbf{x}_a) \tag{1}$$

where $\mathbf{x}_s$ is the smoothed profile, $\mathbf{x}_a$ and $\mathbf{A}$ represents the a priori and averaging kernel for $CH_4$ and $N_2O$ obtained from the ground-based FTIR instrument respectively and $\mathbf{x}_i$ is the initial retrieved profile obtained from satellite measurements after we

interpolated it to the FTIR grid spacing.

The Fioletev method is applied to MIPAS and MLS measurements of $CH_4$ and $H_2O$. It uses variances of the trace gas measurements in latitude bins which can be considered as fairly homogeneous, such that sampling artefacts in the variances can be excluded. We further assume that the measured values of $H_2O$ and $CH_4$ and the errors associated with these measurements are independent. Then the sample variance sigma$^2(M_i)$ of the measurements of one gas by instrument i can be understood as

$$\sigma^2(M_i) = \sigma^2(X_{true}) + \sigma^2(error_i) \tag{2}$$





sigma$^2$(X$_t$rue) is the natural variability of the species under investigation, and sigma$^2$(error$_i$) is the random error of these measurements in terms of variance. Since, putting sampling artefact aside, the natural variability is the same, regardless by which instrument the atmosphere is observed, and since random errors of two independent measurement systems are usually uncorrelated, the variance of the differences between collocated profile measurements by the two instruments depends only on the random errors:

$$\sigma^2(X_{true1} - X_{true2}) + \sigma^2(M_1 - M_2) = \sigma^2(error_1) + \sigma^2(error_2) \tag{3}$$

The terms sigma$^2$(M$_1$-M$_2$), sigma$^2$(M$_1$), and sigma$^2$(M$_2$) are available from the observations, and Eqs (2-3) can be rearranged to give the natural variability and the random error of each of the two measurements.

$$\sigma^2_{natVar} = \frac{1}{2}(\sigma^2(M_1) + \sigma^2(M_2) - \sigma^2(M_1 - M_2))$$
$$\sigma^2_{(M_{1error})} = \frac{1}{2}(\sigma^2(M_1) - \sigma^2(M_2) + \sigma^2(M_1 - M_2)) \tag{4}$$
$$\sigma^2_{(M_{2error})} = \frac{1}{2}(\sigma^2(M_2) - \sigma^2(M_1) + \sigma^2(M_1 - M_2))$$

where $M_1$ represents MIPAS and $M_2$ the MLS or FTIR measurement, depending on the application.

The availability of simultaneous profiles measurements of CH$_4$ and N$_2$O species affords the possibility of internal consistency checks by examining the correlation coefficient and the unexplained variance. For comparison with the FTIR instruments we use only MIPAS measurements at the geolocations of the respective FTIR sites within coincidence criteria of $\pm 2°$ of latitude and $\pm 10°$ of longitude and time difference of $\pm 24$hr. The altitude range of the comparison has been restricted to 18-21 km. For the correlation analysis and the analysis following the Fioletov scheme we use satellite data in latitude bands of 20 degrees.

## 4 Results and Discussion

### 4.1 Bias evaluation

The bias in MIPAS CH$_4$ can be quantified by comparison with the ground based FTIR CH$_4$ at the three selected sites, representing the three latitude bands mentioned. Similarly, the variability of H$_2$O over these three sites has been presented and discussed using the standard deviation of the bias and combined random errors of the instruments between MIPAS and MLS.

The top panel of Fig. 1 shows the results from the comparison between MIPAS CH4 220 and FTIR methane mean profiles. There are 61 coincident measurements at the tropical site, 80 at the mid-latitudinal site and 15 at the high latitude site. The mean relative differences range from 14 % to 3.8 % at altitudes below around 27 km for the tropical site, it is not beyond 10 % in altitude below around 26 km for the mid-latitudinal site. For the polar site the mean relative difference is statistically significant and a positive bias lower than 4.5 % in the altitude range of 18-27 km. We found that the bias is largest for the



tropical site relative to the mid and high latitude deviations, with a positive peak value at tropopause. The average relative differences over altitude are 9.4 %, 5.2 % and 1.6 % along with standard deviation of mean relative differences of 7.7 %, 4.9 % and 4.7 % in altitude ranges of 15-27 km for Addis Ababa, Jungfraujoch and NyÅlesund sites respectively. However, the

5    values reported in Table 1 are averaged over altitude ranges of 15 to 22 km that represents the UT and LS.

The lower panel of Fig. 1 shows the same type of comparison, but for the more recent MIPAS version V5R_CH4_224 over three sites, Addis Ababa, Jungfraujoch and NyÅlesund with a coincident measurements of 29, 17 and 16 respectively. The relative differences range from 4.8 % to -4.6 % at altitudes below around 27 km for Addis Ababa with a positive bias at altitudes below 22 km and a negative bias above 22 km. It is not beyond $\pm$ 4.3 % at altitudes below around 26 km for the Jungfraujoch site, with maximum difference at 26 km. For NyÅlesund, the relative difference is statistically significant and a

positive bias lower than 3.0 % in the altitude range 25-28 km and a negative bias of below -3.8 % at altitudes below 25 km with its maximum difference at around 16 km. We found that the bias is largest for the tropical site, Addis Ababa, relative to the mid and high latitudes with a positive peak value at tropopause. The values reported in Table 2 are averaged over altitude ranges of 15 to 22 km.



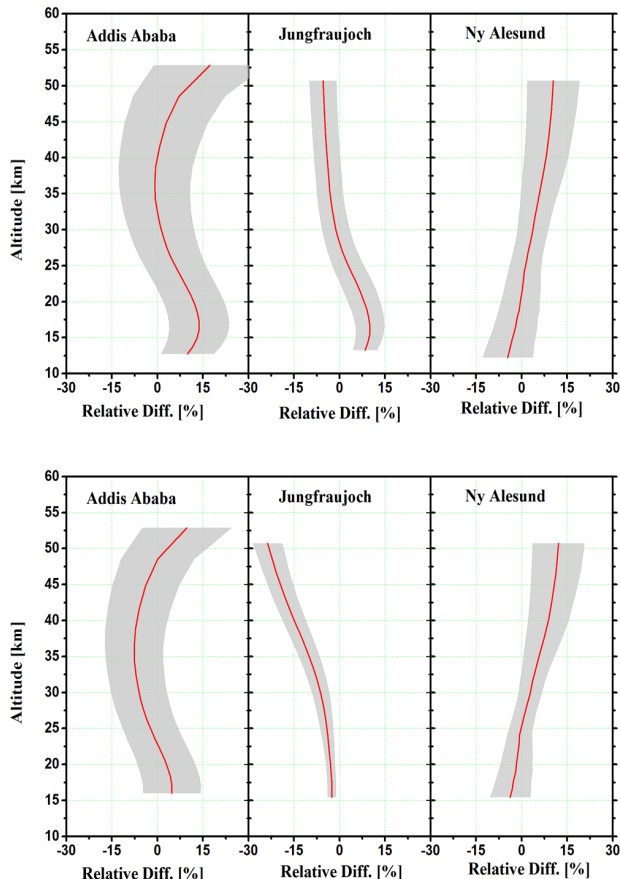

**Figure 1.** Comparisons of MIPAS CH4 220 profile with FTIR values (upper panel) and MIPAS CH4 224 profile with FTIR (lower panel). The mean relative differences for the Addis Ababa, Jungfraujoch and NyÅlesund sites. Shaded area is the Standard deviation of the mean relative differences.

## 4.2 Precision

Fig. 2 shows the scatter plot of MIPAS CH4 220-FTIR (upper panel) and MIPAS CH4 224-FTIR (lower panel). Similar as for the bias, also the scatter is largest in the tropical UT/LS region, compared to the results from the mid-latitudinal and polar site. Further, the data points of the new data version coincide better with the line with unity slope than the data of the older version. The latter tend to lie above this line for altitudes below 22 km for Addid Ababa and Jungfraujoch measurements. This confirms the findings of the previous section, that low altitude methane in V5R_CH4_220 MIPAS data is biased high and that this bias

was reduced in V5R_CH4_224. The standard deviation of mean relative differences between V5R_CH4_220 and FTIR stated in the previous section also shows the precision and it is in the order of 4 to 8 % in altitude between 15 and 27 km of all the three sites and this is comparable to the estimated precision reported by von Clarmann (2006).





The result obtained from the linear fit at the three sites and the root mean squares of residuals between the fitted line and the observations have been summarized in Table 1. Fig. 2 indicates that the uncertainty of MIPAS methane is higher in the tropical UL/LS region than in the UT/LS of mid and high latitudes. This kind of behaviour has also been reported for the operational MIPAS data product provided by ESA (Errera et al., 2016; Payan et al., 2009).

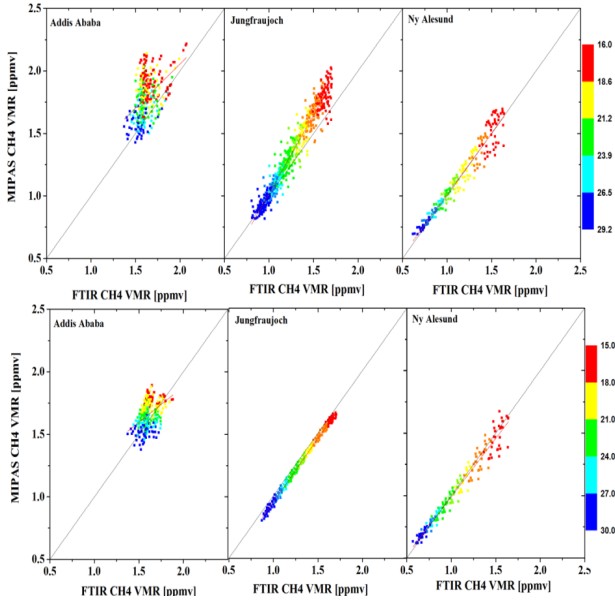

**Figure 2.** Scatter plot of daily mean values of MIPAS CH4 220 (upper panel) and V5R_CH4_220 (lower panel) vs FTIR CH$_4$ of Addid Ababa, Jungfraujoch and NyÅlesund stations from left to right respectively. The colour bar represents the altitude.

**Table 1.** Summary of the comparisons of MIPAS CH4 220 profile with FTIR over the three sites. r is correlation coefficient with in 15 to 22 km.

| Site | Residual | r | Slope | intercept | averaged relative difference ± STD(%) |
|---|---|---|---|---|---|
| Addis Ababa | 5.9 | 0.32 | 0.4 | 1.22 | 12.3± 10.1 |
| Jungfraujoch | 3.6 | 0.84 | 1.2 | -0.07 | 8.4± 5 |
| NyÅlesund | 1.1 | 0.9 | 0.99 | 0.03 | 1.2 ± 6.7 |

MIPAS water vapour observations over the three measurement sites in January 2010 were compared to Aura MLS version 3.3 water vapour. The coincidences criteria were 7° N to 11° N, 45° N to 49° N and 77° N to 81° N and we ascertain 9, 12 and



**Table 2.** Summary of the comparisons of MIPAS CH4 224 and FTIR over the three sites. r is correlation coefficient with in 15 to 22 km.

| Site | Residual | r | Slope | intercept | averaged relative difference $\pm$ STD(%) |
|---|---|---|---|---|---|
| Addis Ababa | 0.4 | 0.18 | 0.13 | 1.5 | 3.9$\pm$ 9.5 |
| Jungfraujoch | 0.04 | 0.98 | 0.97 | 5.3 | -2.6$\pm$ 1.4 |
| NyÅlesund | 0.5 | 0.89 | 0.95 | 0.04 | -2.7 $\pm$ 5.9 |

14 coincidences between MIPAS and Aura/MLS respectively, their temporal coincident is $\pm$ 12 hr. Version 3.3 of Aura/MLS

10 and IMK/IAA MIPAS V5R_H2O_220 data product were used to analyze the precision of MIPAS $H_2O$.

Fig. 3 shows a plot of the standard deviations of mean absolute differences and the combined random errors of instruments as a function of altitude for $H_2O$ vertical profiles obtained from MIPAS and MLS. For the tropical measurements, the combined estimated random error exceeds the standard deviation of the differences inferred from the observations. This indicates an overestimation of the retrieval uncertainties. For the other sites this is not the case although the spatial and temporal coincidence criteria were chosen similar. The result for the tropical site is particularly astonishing because for the combined error we used only the retrieval noise only and ignore the estimates of the other random error components. Thus, and underestimation of the random uncertainty should be expected. However, the spatial and temporal criteria used in all the three atmospheric conditions are similar. Different reasons has reported for the overestimation of standard deviation of the absolute differences, as we have

5 not taking all the sources of errors that contribute to the random uncertainties of the MIPAS measurements (i.e is the only error source considered noise), strong gradients of $H_2O$ spatially and temporally in tropical atmospheric conditions and the overestimation of the standard deviation of mean absolute differences may also be existed due to the natural variability of the parameter ($H_2O$).

### 4.3 Correlation plots of $CH_4$ and $N_2O$

Correlation coefficients of $CH_4$ and $N_2O$ values derived from MIPAS and MLS $CH_4$ are expected not to vary along the latitude. However, both correlation coefficients of V5R_CH4_220 and V5R_N2O_220 of MIPAS as well as MLS version 3.3 $CH_4$ with MIPAS CH4 220 in a global scale with latitudinal bands of 15° and its vertical spacing of 1 km has been found variation at the UT/LS. Thus, the correlation coefficients of below the modest 0.5 found over tropics at upper troposphere and lower stratosphere indicate the presence of large uncertainty of MIPAS CH4 220 (see upper panel of Fig. 4). These results confirm

those of the bias evaluation analysis using the standard deviation of the difference between MIPAS CH4 220 and FTIR methane.

Here in this work the impacts of natural variability of water vapour on the large uncertainty of MIPAS $CH_4$ has been investigated by subtracting the standard variability of water vapour from profiles of MIPAS CH4 220 and MLS methane (see middle panel of Fig. 4). However, the correction coefficients between MLS $CH_4$ (after subtracting $H_2O$ variability) and MIPAS $N_2O$ indicate less than 0.5 over tropics as we are not taking the effects of water vapour variability on $N_2O$. In the last left panel





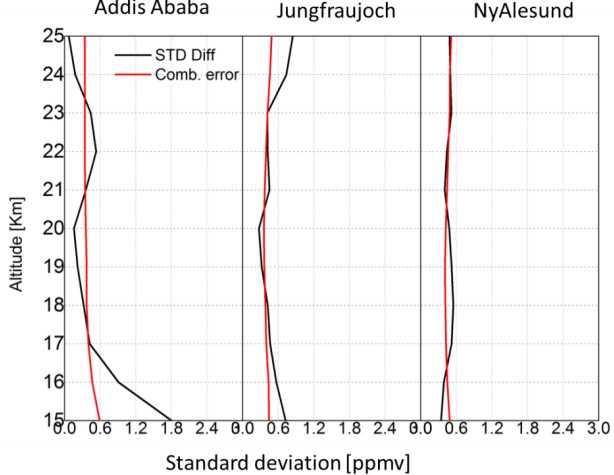

**Figure 3.** The standard deviation of differences versus the combined error of the instruments MIPAS V5R_H2O_220 and MLS measurement for Addid Ababa, Jungfraujoch and NyÅlesund stations (left to right) on January 2010.

of Fig. 4 that has been done after subtracting $H_2O$ variability from MIPAS N2O 220, correlation coefficients of MLS $CH_4$ and MIPAS N2O 220 values are become greater than the modest value 0.5 that clearly shows the impacts of water variability on the uncertainty.

5     The latitudinal variations of correlation coefficients of V5R_CH4_220 and V5R_N2O_220 of MIPAS as well as MLS version 3.3 $CH_4$ with MIPAS CH4 220 have been corrected by taking the natural variability of water vapour in to account and this confirmed the impacts of water variability on the large uncertainty of MIPAS methane existed. As shown in the bottom panel of Fig. 4, The latitudinal variations of correlation coefficients of V5R_CH4_220 and V5R_N2O_220 of MIPAS as well as MLS version 3.3 $CH_4$ with MIPAS CH4 220 have became greater than the modest value 0.5 that indicates the large

10   uncertainty of MIPAS CH4 220 in upper troposphere and lower stratosphere of tropics existed due to the ignorance of water vapour variability impacts on MIPAS CH4 220 data.

    Fig. 5 shows the correlation coefficients between MIPAS CH4 224 and MIPAS N2O 224 as well as with MLS version 3.3 $N_2O$ so that to illustrate the results obtained in MIPAS CH4 220 (see Fig. 4). Moreover, the correlation coefficients shown as a function of latitude did not vary as shown in MIPAS CH4 220 as its values are greater than the modest value. Here, we can conclude that the uncertainty of MIPAS CH4 224 has been reduced as compare to MIPAS CH4 220 that is in agreement to the result reported by (Plieninger et al., 2015, 2016). The impacts of water vapour on the new version MIPAS CH4 224 data has been reduced as water vapour that is the interference gas has also been retrieved with methane simultaneously.



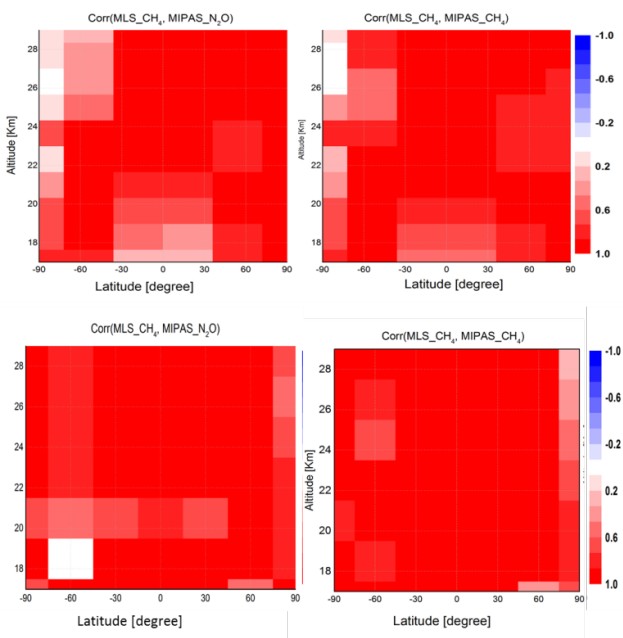

**Figure 4.** Correlation coefficients of MLS $CH_4$ and MIPAS N2O 220 (left) and MLS $CH_4$ and MIPAS MIPAS CH4 220 (right) as a function of latitude and MIPAS altitude for Feb., 2010 before subtracting water vapour variability (Top panel), after subtracting water vapour variability from MIPAS N2O 220 and MIPAS N2O 220 (bottom panel).

### 4.4    $H_2O$ Variability Versus MIPAS $CH_4$ Uncertainties

For atmospheric constituent measurements, the sample variance includes the natural variability of the measured quantities in addition to the variance of random error. Natural variability of water vapour over three sites can be derived from the data sets provided by at least two or more different techniques. Using Eq. 4, natural variability in terms of standard variation of water vapour over Addis Ababa was determined using January 2010 data set from MIPAS and MLS. For the same purpose we use collected MIPAS and FTIR (Addis Ababa) water vapour data sets measured between May 2009 and February 2011. The

uncertainty of MIPAS methane over Addis Ababa is investigated using the two satellite measurements to determine the natural variability of water vapour as well as the uncertainty of MIPAS measurements of $CH_4$.

The random uncertainties of the measurements can be estimated from sets of pairs of collected data from MIPAS and FTIR or MIPAS and MLS as described in the methodology section. Fig. 6 shows the estimated precisions of the MIPAS CH4 220 and V5R_CH4_224 products for the three sites. For V5R_CH4_220 the mean inferred uncertainties in the altitude range between 15 and 27 km are 5.9 %, 4.8 %, 4.7 % for Addis Ababa (black solid line), Jungfraujoch (red solid line) and NyAlesund (green solid line) respectively. These results were obtained by application of Eq. 4 to the FTIR data from the three stations and collected MIPAS measurements. These MIPAS random uncertainties are larger for Addis Ababa collections than for Jungfraujoch and





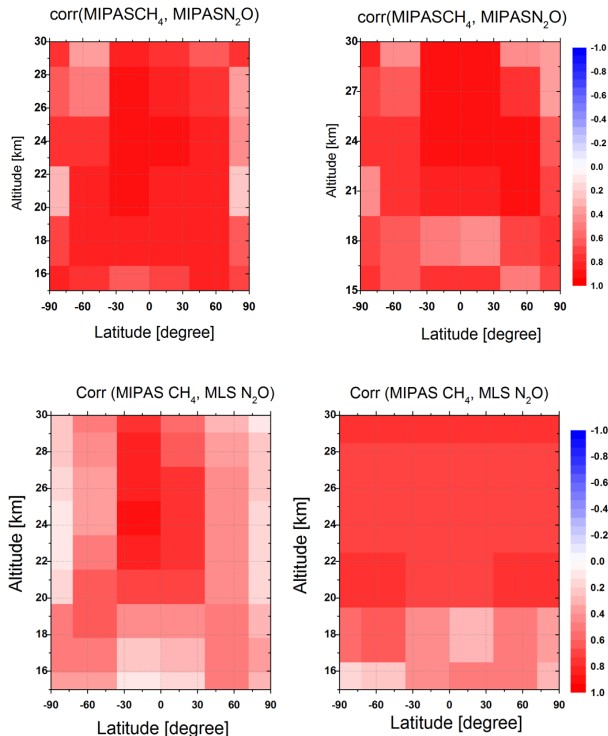

**Figure 5.** Correlation coefficients of MIPAS CH4 224 and MIPAS MIPAS N2O 224 (top) and MIPAS CH4 224 and MLS $N_2O$ (bottom) as a function of latitude and MIPAS altitude for Feb., 2010 before subtract water vapour variability (left column) and after subtracting water vapour variability from both $CH_4$ and $N_2O$ (right column).

NyAlesund. However, the inferred uncertainties have become smaller for the MIPAS CH4 224 data version (Fig. 6, right panel). They are 2.4 %, 1.4 %, 5.1 % for the three sites.

Fig. 7 shows the natural variability of MIPAS V5R_H2O_220 determined using Eq. 4 applied to MIPAS V5R_H2O_220 and MLS (Ver3.3). The resulting natural variabilities at 15-17 km altitude are 8.4 %, 5.4 % and 3.4 % for Addis Ababa, Jungfraujoch, and NyAlesund.

## 5 The role of water variability on the uncertainty of MIPAS $CH_4$ in tropics

In this part of the analysis the role of water vapour variability in the tropical tropospheric layer in the uncertainty of MI-PAS methane is explored using correlation analysis. Temporal variability of water vapour in the upper troposphere and lower stratosphere (121-56 hpa) of tropics has been determined from MIPAS H2O 220 for the time period of Jan, 2009 to Apr, 2012.

The role of LS water vapour variability on the TTL uncertainty of MIPAS methane can be investigated using monthly variability of water vapour over the tropical tropopause layer and comparing with the monthly random uncertainty of MIPAS





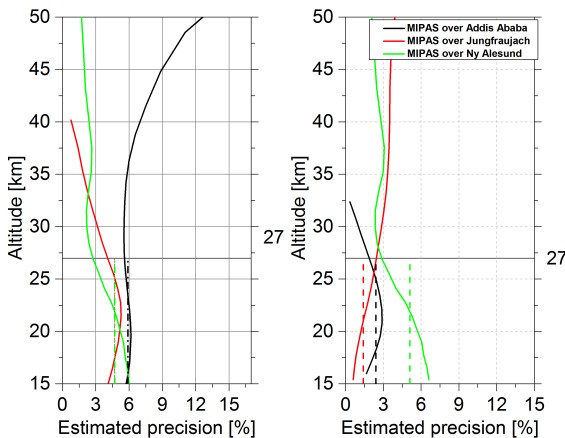

**Figure 6.** Estimated random uncertainty of MIPAS CH4 220 (left) and MIPAS CH4 224 (right) from FTIR comparison over the three atmospheric conditions.

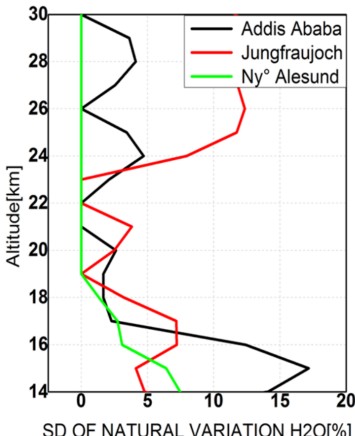

**Figure 7.** The natural variability of $H_2O$ over three measurement sites, computed from MIPAS and MLS.

methane. The colour bar indicates the correlation coefficients between the two variables on temporal resolutions of a month and altitudinal spaces of 1 km in altitude ranges of upper troposphere and lower stratosphere of tropics using the three year (2009-2011) MIPAS $CH_4$ and $H_2O$.

Table 4 shows the seasonal variations of both water variability and uncertainty in winter and spring at different layer of the atmosphere of the three sites. Both water variability and uncertainty of MIPAS $CH_4$ are high in tropics throughout the year.





**Table 3.** Standard uncertainties of MIPAS CH4 220 and standard deviations of seasonal water vapour variability over Addis Ababa, Jungfraujoch and NyÅlesund.

| layer | H2O var (%) | | | | | | SD Uncert MIPAS CH4(%) | | | | | |
|---|---|---|---|---|---|---|---|---|---|---|---|---|
| | Tro | Mid | Pol | Tro | Mid | Pol | Tro | Mid | Pol | Tro | Mid | Pol |
| 15-17 km | 24.2 | 8.2 | 5.6 | 19.8 | 11 | 4.5 | 7.2 | 4.0 | 4.4 | 7.5 | 5.6 | 4.8 |
| 18-21 km | 9.5 | 5.8 | 7.3 | 7.3 | 4.9 | 4.5 | 5.3 | 3.9 | 5.0 | 5.3 | 3.8 | 3.5 |
| 22-25 km | 5.9 | 9.3 | 13.7 | 5.4 | 4.9 | 6.3 | 3.6 | 4.7 | 7.9 | 3.5 | 4.1 | 4.3 |

**Table 4.** Standard uncertainties of MIPAS CH4 224 and standard deviations of seasonal water vapour variability over Addis Ababa, Jungfraujoch and NyÅlesund.

| layer | H2O var (%) | | | | | | SD Uncert MIPAS CH4(%) | | | | | |
|---|---|---|---|---|---|---|---|---|---|---|---|---|
| | Winter | | | Summer | | | Winter | | | Summer | | |
| | Tro | Mid | Pol | Tro | Mid | Pol | Tro | Mid | Pol | Tro | Mid | Pol |
| 15-17 km | 24.2 | 8.2 | 5.6 | 19.8 | 11 | 4.5 | 7.8 | 4.7 | 5.6 | 7.7 | 4.1 | 3.7 |
| 18-21 km | 9.5 | 5.8 | 7.3 | 7.3 | 4.9 | 4.5 | 5.1 | 3.9 | 4.9 | 4.6 | 3.5 | 3.4 |
| 22-25 km | 5.9 | 9.3 | 13.7 | 5.4 | 4.9 | 6.3 | 2.9 | 3.9 | 5.6 | 2.8 | 3.1 | 3.0 |

Mainly in the lower stratosphere (18-21 km) both the variability of water and the uncertainty of MIPAS are higher in tropics as compared to other latitude bands.

We can explore the relationship between the random uncertainty of MIPAS methane and the natural variability of water vapour by employing F statistics and correlation analysis method. Fig. 8 is a scatter plot of the standard natural variability of water vapour and the standard random uncertainty of MIPAS methane for the lower stratosphere over tropics using monthly
averaged values for time period of three years (Jan., 2009- Dec., 2011).

The large uncertainty of MIPAS CH4 220 at Addis Ababa site is correlated or coupled with water variability. The correlation of the monthly standard natural variability of water vapour and standard uncertainty of MIPAS methane over Addis Ababa at lower stratospheric layer (18-21 km) is 0.88. The variability of water at the TTL is large which might have an association to the large uncertainty of MIPAS CH4 220 for tropical atmospheric conditions, as the production of $H_2O$ in the lower stratosphere
has also through oxidation of methane in the upper troposphere and transport processes. The variability of $H_2O$ in TTL is highly correlated to the temperature at the tropopause (Takele Kenea, 2014). However, the correlation between MIPAS CH4 224 uncertainty and water vapour variability became reduced as water vapour was jointly fitted with $CH_4$ (see left panel of Fig. 8).

Left panel of Fig. 8 shows the association large uncertainty of MIPAS CH4 220 with water variability in altitude range
5  between 18 and 21 km and the right panel is for MIPAS CH4 224. However, the correlation coefficients that indicate the effects





**Table 5.** Summary of correlation analysis of water vapour variability with random uncertainty of MIPAS CH4 220 (first row) and MIPAS CH4 224 (second row) from monthly averaged values in the lower stratosphere.

| Residual | CC | $R^2$ | intercept | slope | P value |
|----------|------|------|-------------------|-------------|---------|
| 0.01 | 0.88 | 0.78 | $0.002 \pm 0.004$ | $0.82 \pm 0.04$ | 0.001 |
| 0.07 | 0.95 | 0.72 | $0.8 \pm 0.02$ | $2.7 \pm 0.3$ | 0.001 |

of water variability on the uncertainty of MIPAS CH4 of both version (220 and 224). In the tropics the impacts of water vapour on the new version data of MIPAS CH4 224 has been reduced as their correlation coefficients in that altitude range is also less.

We conclude that the variability of water had an impact on the large uncertainty of MIPAS $CH_4$ measurements in tropical atmospheric conditions. MIPAS $CH_4$ shows larger uncertainty for the tropical than for midlatitudinal and polar site.

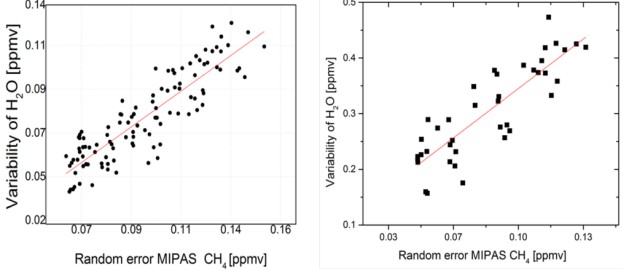

**Figure 8.** The natural variability of $H_2O$ and the random uncertainty of MIPAS CH4 220 (right) and MIPAS CH4 224 (left). Using a three years data sets, 2009-2011 for altitude 18-21 km of tropics in the lower stratosphere.



## 6 Summary and conclusions

In this paper, we apply different methods to investigate the large uncertainty of MIPAS methane in the tropical upper tropo-
sphere and lower stratosphere as compared to the uncertainty at the same altitude level of mid and high latitudes. Similarly,
the latitudinal variation of standard natural variability of water vapour has been determined using those techniques. Finally,
we detected the natural variability of water vapour was a cause for high uncertainties on the MIPAS CH4 220 measurements
over tropics using the correlation coefficient analysis and F statistical analysis. This causal has reduced in the new data version
MIPAS CH4 224, where $H_2O$ was jointly fitted with the target gas, $CH_4$.

Mean relative differences of -12.3 %, 8.4 % and 1.2 % were obtained between MIPAS CH4 220 and ground based FTIR
for the three sites in altitude range 15-21 km. A latitudinal dependence of MIPAS $CH_4$ uncertainty is found. Moreover, the
average estimated uncertainties of MIPAS CH4 220 were obtained 5.9 %, 4.8 % and 4.7 % using the differential method in
altitude ranges of 15 to 27 km for tropics, mid and high latitudes, respectively. The estimated measurement uncertainties of
MIPAS CH4 220 are different for the three sites, representing tropical, midlatitude and polar conditions. This confirms the
latitude-dependence of MIPAS methane uncertainties. The correlation coefficient between two long lived species, MIPAS CH4
220 and V5R_N2O_220 also hints at a large uncertainty of lower stratospheric MIPAS CH4 220 over Addis Ababa, as R is less
than 0.5. At this site, also this correlation coefficient is only around this modest value of 0.5 at the other altitude levels. This
indicates the latitudinal variation of MIPAS CH4 220 uncertainties at lower stratosphere over tropics is larger than the same
levels of mid and high latitudinal bands.

Compared to the older data version, the mean relative differences between MIPAS and FTIR at 15-21 km altitude have been
considerably reduced for data version V5R_CH4_224. They are now 3.9 %, -2.6 % and -2.7 % for tropical, midlatitudinal
and polar site, respectively. Moreover, also the average estimated uncertainty at 15-27 km altitude inferred by the differential
analysis has became smaller with the new MIPAS data version. They are now 2.4 %, 1.4 % and 5.1 % for the sites under
assessment. While the magnitude of uncertainties became significantly smaller, their latitudinal dependence is still there.

Similarly, the intercomparison results of water vapour derived from MIPAS and MLS for the three atmospheric conditions,
the standard deviation of the mean difference is larger than the combined random errors in all the three conditions. However
the variation is large in the tropics that is five times that of the combined error in the altitude below 17 km and this indicates
the natural variability of water vapor is high in upper tropospheric layers of tropics.

Therefore, the uncertainty on MIPAS CH4 224 has been reduced as compared to the old version, MIPAS CH4 220. The
reason for the reduction of its uncertainty is the retrieval approach employed during retrieval of $CH_4$ and $N_2O$. The contribution
of natural variability of water vapour on the uncertainty of MIPAS CH4 224 has been reduced as its main interfering gas water
vapour has been fitted jointly with $CH_4$ and $N_2O$ (Plieninger et al., 2016).



**acknowledgements**

. We greatly acknowledge the MLS science teams for the satellite data used in this study. We also acknowledge two FTIR data providers used in this publication that were obtained as part of the Network for the Detection of Atmospheric Composition Change (NDACC) and are publicly available (see http://www.ndacc.org). Special thanks go to Dr. samuel takele for his contribution on calibrating the spectra measured by the FTIR at Addis Ababa. Finally, authors would like to thank Mekelle and Addis Ababa universities for the sponsorship and financial supports.



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
