# Peer review of "Impacts of H2O variability on accuracy of CH4 observations from MIPAS satellite over tropics"

_Atmospheric Measurement Techniques, 2019_

## Referee Comment (RC1) · Anonymous Referee #1 · 7 Jun 2019

**1   General Comments**

This paper examines the variability in MIPAS CH$_4$ data, ascribing some of its variability to interference from water vapour and noting that this problem with the data has been at least partly corrected in the current release. The paper reports useful progress in the understanding of the quality of this important dataset. It should therefore be published. However, I am of the opinion that it requires a great deal of revision first.

Some of the work which the paper requires is merely technical corrections. The standard of written English, referencing and technical typesetting is patchy; I make many

suggestions for improvement below, but these are by no means exhaustive; it is not the task of the reviewer to do a complete copy-edit on the paper. And there are many ways in which the figures could be improved.

The main issue that I have with the paper is more structural: it needs to be clearer and more explicit about its aims and about how (and whether) these have been achieved. The paper feels like a collection of plots that have been produced while investigating a data set, which have all been thrown into a document without sufficient thought as to which figures are really needed to explain the point the authors are trying to make. To be published, it really needs a thorough re-organisation, and the authors need to explain what they did in some more detail and far more clearly.

**2 Specific comments**

- Page 1 line 11 "Moreover, the correlation coefficient between MIPAS CH4 220 and MIPAS `V5R_N2O_220` is 0.32 in the upper troposphere and lower stratosphere over tropics and larger than the modest value 0.5 in mid and high latitudes." This is odd wording; it is not clear what the specific relevance is of the value 0.5, nor how much greater than 0.5 the correlation coefficient is in mid/high latitudes.

- Page 5 line 18 : "The altitude range of the comparison [between MIPAS and FTIR] has been restricted to 18-21 km". Does it make any sense to do that, given that the vertical resolution of a FTIR instrument is far worse than 3 km?

- Page 8 figure 2: This figure shows nicely how $CH_4$ version 224 agrees better with FTIR than $CH_4$ version 220. However, there are a number of fixes needed

    – The text on the figure may be a little small, especially on the colour bar. Text on a figure should end up about the same size as the caption text in the final

version of the paper.

- – The colour bar does not match the colours in the dots. In particular, the dots include an orange colour which does not appear in the colour bar.
- – The cyan and green colours in the colour bar are not easy for the eye to distinguish. The yellow is very distinct from the other colours, but can be hard to see on a white background. Selecting a suitable colour scale is not trivial, and one should never be satisfied with the default colours in a plotting package.
- – The caption says that both upper and lower panels are version 220. It is clear from the text that the lower panels are version 224, so the caption needs correcting.
- – The altitudes in the colour bar are clearly in km; the caption should say so.
- – There is a linear fit line of some sort shown in the figure, but it is difficult to see and the caption does not say what it is for. Either the line should be made clearer and the caption should explain its purpose, or the line should be removed.
- – In plots such as this, where the two axes are for two estimates of the same quantity, and in the same units, the scale of the axes should be such that the 1:1 line is at $45°$ to the axes.
- – It is good practice to add labels (a), (b) etc. to panels of a multi-panel figure so that they can be referred to from the caption.

- Page 8, last two lines, running on to page 9: It seems very odd to do such a limited comparison of MIPAS and MLS $H_2O$. Either the two instruments are known to agree well (in which case, why do a comparison at all?) or they are known to differ (in which case, how much can we learn from such a limited comparison?).

- Page 9 Lines 10–14: I am struggling to understand all this from the explanation given. We can see from figure 2 that $CH_4$ has only a very small range of possible

values at a given height in the tropical lower stratosphere. This will also be true of any chemically-stable tropospheric source gas, including $N_2O$. The measurements could be unbiased and with relatively small random measurement error and you would still find them to have little correlation in a small altitude/latitude range. So the low correlations do not necessarily indicate large uncertainty, they may just indicate small true variability in both species. (Of course, both species decrease with height, so the correlation between them will appear good if you include data from a wide range of altitudes.)

- Page 9 lines 16–20: I do not understand what is meant by "subtracting the standard variability of water vapour from profiles of MIPAS CH4 220 and MLS methane" The authors need to explain what they did in more detail. A reader is supposed to be able (at least, in principle) to go away and repeat the work in a paper for himself. Based on this description, I would not be able to do so.

- Page 11 figure 4: If the authors decide to retain this figure, there are several improvements that could be made:

  – The title above each panel should be removed.
  – It is good practice to add labels (a), (b) etc. to panels of a multi-panel figure so that they can be referred to from the caption.
  – In general it is good practice to use a diverging (two-sided) colour scale such as that used here for a quantity which tends to be equally spaced about zero. But in this case there are no cells which have a negative correlation. The figure would therefore be clearer if a good single-sided scale were used, showing values only between zero and one. The caption could confirm that there are no negative correlations.

- Page 11 lines 7–9: I am concerned as to the validity of the results obtained by this procedure, given the very small sample size which the authors note earlier in

the paper. Why not use the MLS and MIPAS data over the full time for which the FTIR data was available?

- Page 13 figure 6: It is mystifying to me that the red curve in the left-hand panel and the black curve in the right-hand panel stop at 40 and 32 km respectively. Also, the caption needs to explain what the dashed lines between 15 and 27 km represent.

- Page 12 lines 6-8 and page 13 figure 7: The text says that the figure shows the "natural variability" of the MIPAS $H_2O$ — I am not clear on what this means. But if this figure is obtained from equations (4), as I rather infer, then it shows the actual natural variability of $H_2O$, inferred from both the MIPAS and MLS data. The authors need to me much more careful to ensure that their writing is clear and unambiguous.

- Page 13 line 15: The sentence refers to a colour bar, but does not say in which figure this colour bar is to be found. The section is discussing $H_2O$ — $CH_4$ correlations, but there is no figure in the paper showing such a correlation with a colour scale.

- Page 14 lines 7–10 and page 15 figure 8: It is not stated whether this figure is for a single location, an area of the globe, the entire world. Furthermore, it is not explained why one figure has many more points on it than the other (about 106 vs about 39). As the data are stated to be monthly data for three years, 39 points is approximately correct, 106 points is clearly wrong.

**3  Technical corrections**

- Page 1 Lines 9-11 and elsewhere: Is it usual to put a space between a number and the % sign? If it is, it should be a non-breaking space (\, or ~ in LaTeX) in

order to ensure that the % is on the same line as the number.

- Page 2 Lines 5-6: It is not good style to start two consecutive sentences with "However, . . . ".

- Page 2 line 14: "on the upper troposphere" should be "in the upper troposphere".

- Page 2 Line 22: "been improve" should be "been improved"

- Page 2 line 25-26: This sentence repeats itself and should be shortened.

- Page 2 line 28: "revealed in (Payan et al., 2009; Errera et al., 2016)" should be EITHER "revealed (Payan et al., 2009; Errera et al., 2016)" OR "revealed in Payan et al., (2009) and in Errera et al., (2016)". Elsewhere in the paper, the authors should also be careful when citing a paper as to whether the author's names are part of their sentence (in which case only the year is in parentheses) or whether the author's name is not part of the sentence (in which case both the name and year are in parentheses).

- page 2 line 31: "did not quantified" should be "did not quantify".

- page 3 lines 2–4: This sentence is quite long and involved. The points could be made in a clearer and less ambiguous way, possibly with more, and shorter, sentences.

- Page 3 line 5: "sections 3" should be "section 3".

- Page 3 lines 9–11: This sentence is badly worded, in particular, "along with" should simply read "with".

- Page 3 lines 24–25: Another example of wrongly-placed brackets in a citation. "Lambert et al. (2007)" should be "(Lambert et al. 2007)".
- Page 3 line 26: ...and another: "Minschwaner et al. (2015)" should be "(Minschwaner et al., 2015)".

- Page 3 Line 26 "The vertical resolution of MLS CH$_4$ is between 4 and 5 km and 4 to 6 km for N$_2$O". This is another example of the odd wording prevalent in this paper. The sentence would be better worded as "The vertical resolution of MLS CH$_4$ is between 4 and 5 km; the vertical resolution of N$_2$O is 4 to 6 km.". While I am looking at this paragraph, why do the authors state the vertical resolution of CH$_4$ and N$_2$O, but not for H$_2$O?

- Page 4 line 6 and elsewhere: "NyÅlesund" should be "Ny-Ålesund"; it is two words with a hyphen between them.

- Page 4 Line 13 and many places elsewhere: "CH4" should be "CH$_4$". The only exceptions should be where the symbols form part of the name of a MIPAS product, e.g. V5R_CH4_224

- Page 4 line 29: sigma$^2$ should be $\sigma^2$ and the letter $M$ should be in italics.

- Page 4 line 30 It is better style to define some symbols in order to avoid using words such as *true* and *error* in equations. If you must use words, then they should be in non-italic type. So you could go for

$$\sigma^2(M_i) = \sigma^2(X_\text{true}) + \sigma^2(\text{error}_i)$$

but it is preferable to write:

$$\sigma^2(M_i) = \sigma^2(X_t) + \sigma^2(\epsilon_i)$$

where $X_t$ is the true value of the measured quantity and $\epsilon_i$ is the measurement error of the $i$-th instrument.

- page 5 lines 3, 9 and 14: The mathematical symbols in the text need fixing.

- page 5 line 11: Again, it is better to define some symbols and then use them than to have long words in the equations. Everything to the right of the = signs is fine, but the various $\sigma^2$s to the left of the = signs could be improved a great deal.

- Page 5 line 18, and probably many other places: A non-breaking space should be used between a number and its unit in order to avoid the number being at the end of one line and the unit at the start of the next. In LaTeX, use $\backslash$, between a number and a unit to get a thinner space. Use ~ between "figure" and its number to get a normal-sized space. In word processors, use Ctrl-Shift-Space for a non-breaking space.

- Page 8 table 1: Is it not usual to use $R$ for a correlation coefficient? Whatever symbol is used it is a symbol, so it should be in italics.

- Page 10 figure 3: In the caption "Addid Ababa" should be "Addis Ababa". Also, the range 1.8–3.0 ppmv appears on the plot, but is not used. The plot would be clearer if the axis went from 0.0 ppmv to 2.0 ppmv.

- Page 11 line 7: "at least two or more" should be "at least two". You do not need the "or more".

- Page 13 figure 6: Remove the words "MIPAS over" from the caption.

- Page 13, figure 7: The letter Åhas not reproduced correctly in the legend.

- Page 16 Line 24: "R" is a symbol so it should be in italics.

- Page 17 Line 7: Samuel Takele's name should have capital letters.

- Page 18 line 20: All ACP papers have a DOI — references should all include a DOI where it is available.

- Page 19 line 35: Not a complete reference, and part of it is incorrectly in ALL CAPITALS.

- Page 19 line 37: The authors should avoid referencing grey literature like this.

- Page 20 line 16: Some xml or html tags have crept into the reference.

- Page 20 (line numbering gone mad in draft) The reference to Randel and Jensen (2013) is not done correctly.

---

## Referee Comment (RC2) · Anonymous Referee #2 · 12 Jul 2019

Paper review

This paper shows that water vapor variability impacted the methane concentration retrievals from MIPAS 220 and is improved in the 224 version. The paper uses a methodology of using coincident measurements from two data sources to derive the atmospheric variability and each intrument's random noise contribution following a technique published by Fioletov 2006. First here are some major issues I have with the paper. Given the basic assumptions the Fioletov method I believe is OK however I think the authors need to consider some possible limitations. One being that the instrument noise may depend on the concentration amount of methane due to forward model non

linearities. This can be checked by correlating the retrieved amount against its reported uncertainty supplied by the MIPAS team. Secondly when deriving the instrument noise estimate from the coincident data, it would be interesting to compare that to the value supplied with the data set as a validation of the method. Neither of these were not done.

The presentation is not clear in many places and needs to be reworked a lot in order to publish this. Figure 4 for example shows a before and after like correlation analysis for methane version 220. In the after figure the authors say they subtracted water vapor variability from the CH4 retrieval and show how much better the corrrelation has improved. I have no idea how you can after the fact remove the water vapor variabilty impacts from the 220 data set or the 224 (figure 5) data set (which itself is significantly improved in this regard due to the simultaneous retrieval of H2O and CH4). Because the authors provide no explaination of how this is done, I am recommending rejection. The "removing" the effect of H2O interference leading to greatly improved agreement with correlative data is the central point of the paper and establishment of cause described in the title, this needs to be much better explained.

The text was hard to follow and comprehend and I give a few examples of this below.

On page 9 line 18 there is a reference to a middle panel in figure 4 a figure with 4 panels in a 2X2 arrangement. What is the middle panel?

Figure 7 show profiles of H2O variability at three station sites. The profile at some altitudes is clipped at zero suggesting that it either is negative (not possible) or unknown. One profile it is exactly zero which is extremely unlikely (ie absolutely no atmsopheric variability).

I dont know why the figure 8 scatter plot of monthly averages in one height range for the 224 data set contains many more the 36 points for 3 year monthly averages.

A sentence on page 13, line 15 seems to refer to a figure not included in the paper. It

cannot possibly be describing figure 8.

There are a lot wording and grammer errors in here that need improving.

---

## Referee Comment (RC3) · Anonymous Referee #3 · 22 Aug 2019

General Comments

This manuscript presents an investigation of the large uncertainty of MIPAS V5R_CH4_220 methane retrievals in the tropical upper troposphere and lower stratosphere and attributes it to interference from atmospheric water vapour. Including water vapour when fitting methane reduces the uncertainties in the new data version MIPAS V5R_CH4_224.

The work is a useful contribution to the field and appropriate for AMT, but the manuscript needs major revisions and another round of reviews.

I have read the reviews of the other two referees. My assessment of the manuscript is

similar to theirs, so I will not repeat their specific and technical comments.

In general, the manuscript should be improved in the following ways:

- Change the title. MIPAS is not a satellite. e.g., "The impact of $H_2O$ variability on the accuracy of MIPAS $CH_4$ measurements [or retrievals] over the tropics:"

- Rewrite the abstract for clarity, conciseness, and grammar. The explanation of the results is wordy and unclear.

- The manuscript needs much clearer explanations of methods and results throughout, and more detail on how results were obtained.

- Figures need to better describe what is shown and ALL plots and captions need revisions. e.g., Figure 1 shows (X - Y)/Z differences but doesn't say what X, Y, and Z are. Figures 4 needs a better colour scale, panel labels, y-axis label, larger fonts, etc. Figure 8 should plot $CH_4$ vs. $H_2O$, not $H_2O$ vs. $CH_4$. Take a careful look at quality of all the figures.

- Is it correct to extrapolate the results from three specific sites as representative of the tropics, mid-latitudes, and polar regions? Justify this assumption.

- Correlation seems be equated with cause, e.g., p16, para1.

- Add a table stating sites/latitude bands used and time periods.

- Use consistent terminology when referring to the MIPAS versions, e.g., both V5R_CH4_220 and MIPAS CH4 220 are used. Use the former throughout, and similarly for V5R_CH4_224.

- Section 4.3 should be rewritten for clarity.

- Data providers who are co-authors don't need to be thanked in the Acknowledgements.

- The References should be revised to ensure that they are correct and have consistent

formatting. Some have incomplete information and several are old AMTD references (e.g., Laeng et al., 2015; Sepulveda et al., 2012). AMTD references are to manuscripts under review. These should be updated to the published AMT references.

- The manuscript needs line-by-line copy editing to correct the many typographical, grammatical, and technical errors.

- Overall, the manuscript is poorly written. It needs a complete rewrite for scientific clarity. I encourage all of the authors to review the next version carefully prior to resubmission.

---

## Author Comment (AC1) · 19 Sep 2019

Responses to referee 1: (received: 7 June 2019)

We thank the referee for very insightful questions and comments. They have helped to improve the quality of the paper. We have substantially revised and reorganized the manuscript, in many parts extended sentences and paragraphs has added. Our responses are given point-by-point below (blue Times New Roman font) following each of the reviewers' comments, which are repeated in full (black Times New Roman Italic font). Reproduced text from the revised manuscript is set in green Times New Roman font.

In addition to the change made in the manuscript to take into account your comments or the comments of the other referees, several other changes have been made and are listed here. The page number and lines indicated on the lists are taken from commented manuscript.

**1. General Comments**

This paper examines the variability in MIPAS CH4 data, ascribing some of its variability to interference from water vapour and noting that this problem with the data has been at least partly corrected in the current release. The paper reports useful progress in the understanding of the quality of this important dataset. It should therefore be published. However, I am of the opinion that it requires a great deal of revision first.

Some of the work which the paper requires is merely technical corrections. The standard of written English, referencing and technical typesetting is patchy; I make many suggestions for improvement below, but these are by no means exhaustive; it is not the task of the reviewer to do a complete copy-edit on the paper. And there are many ways in which the figures could be improved.

The main issue that I have with the paper is more structural: it needs to be clearer and more explicit about its aims and about how (and whether) these have been achieved. The paper feels like a collection of plots that have been produced while investigating a data set, which have all been thrown into a document without sufficient thought as to which figures are really needed to explain the point the authors are trying to make. To be published, it really needs a thorough re-organisation, and the authors need to explain what they did in some more detail and far more clearly.

Response: We would like to thank the reviewer for this positive evaluation and critical comments that would help us to make the paper more vital to the scientific community. Furthermore, the referee was also commenting on the technical corrections that should be included in the revised manuscript. This paper has addressed the cause for the large uncertainty in the old version of MIPAS\_CH4\_220 concentration at the upper troposphere and lower stratosphere of tropics through exploring the effect of water vapour variability on the uncertainty. However, the uncertainty of MIPAS\_CH4\_224 has reduced as water profile has retrieved jointly with methane.

The manuscript has been clarified. See also our responses to your specific comments and technical corrections below.

As the referee suggested on the structure of the manuscript, we have re-organized the manuscript in a way that readers can easy understand by adding detail explanation of methods and results as well as new subsections under result and discussion section. The new subsection is also recommended by other referees.

We inserted at the end of P1L2: "at upper troposphere and lower stratosphere."

P3L2-5: the sentences have been replaced by "The coincident measurements Of  $H_2O$ ,  $CH_4$  and  $N_2O$  by MIPAS, ground based FTIR and MLS were used to estimate the uncertainty of MIPAS\_CH4\_220 and MIPAS\_CH4\_224 profiles and the natural variability of  $H_2O$ . MLS CH4 was derived from EOS MLS coincident measurements of atmospheric water vapour ( $H_2O$ ), carbon monoxide (CO) and nitrous oxide ( $N_2O$ )."

Inserted after the period in P3L4; Different methods has applied to determine uncertainty of MIPAS\_CH4\_220, MIPAS\_CH4\_224 measurements and variability of water vapour at the three latitudinal bands. Intercomparison results of methane (CH4) measured by MIPAS with the ground based FTIR products obtained from Addis Ababa FTIR observatory and other two NDACC FTIR sites (Jungfraujoch, Switzerland and Ny-Ålesund, Spitsbergen). It has been analyzed using the statistical analysis methods detailed in von Clarmann (2006). Natural variability of water vapour and uncertainties of MIPAS methane can also be determined using differential method proposed by Fioletov et al. (2006) and applied on different literatures (Toohey et al. (2007); Sofieva et al. (2014) for the three atmospheric conditions using at least two different measurement techniques. Furthermore, correlations analysis between.CH4-N2O measured by MIPAS and MIPAS CH4 as a function of latitude and altitude in a global scale. Finally, the cause of high uncertainty of MIPAS\_CH4\_220 and its reduction in MIPAS\_CH4\_224 at the lower stratosphere of tropics has been assessed through taking its relation with water vapour variability using a regression analysis method.

Inserted after the period in P5L14; "Both the estimated standard deviation (SD) of instrument uncertainty (i.e. MIPAS  $CH_4$ ) and standard deviation of water variability for a given location, time of year, and layer were obtained using equations 4. Applying equation (4) to these data sources creates two sets of SD of MIPAS  $CH_4$  uncertainty estimates. Similarly, SD of water vapour variability was obtained for each of the three latitudinal bands. The value estimated SD uncertainty of MIPAS CH4 was calculated as square root of the mean variance estimates from the two data sources."

The following paragraph has been added as a last paragraph under methodology section so that to make clear the methods employed in the manuscript.

Replace the last paragraph in P5L15-20 by "In addition to the above methods employed in

this paper, as the UT/LS, mixing ratios of these long-lived trace gases are largely controlled by dynamical processes, generally resulting in compact tracer-tracer correlations. These correlations are usually more compact in high and mid-latitudes, while in tropics a somewhat larger scatter is observed (Plumb et al., 2007; Payan et al., 2009). We used such methods to show the variation of MIPAS\_CH4\_220 uncertainty with high value at LS of tropics and its reduction in MIPAS\_CH4\_224 as a function of latitude and altitude in a global scale using corresponding values MIPAS\_N2O\_220, MIPAS\_N2O\_224 for February 2010. In addition, both version data sets of MIPAS CH4 and MLS CH4 version 3.3 for February 2010 have been discussed too. These correlations are calculated on latitude bins space by 30o and on an altitude grid with 7 levels and spacing of 2 km."

**2. Specific comments**

• Page 1 line 11 "Moreover, the correlation coefficient between MIPAS CH4 220 and MIPAS V5R\_N2O\_220 is 0.32 in the upper troposphere and lower stratosphere over tropics and larger than the modest value 0.5 in mid and high latitudes." This is odd wording; it is not clear what the specific relevance is of the value 0.5, nor how much greater than 0.5 the correlation coefficient is in mid/high latitudes.

Response: We agree that the exact values in mid and high latitudes have to be included in the text.

Page 1 line 11: Moreover, the correlation coefficient between MIPAS\_CH4\_220 and MIPAS\_N2O\_220 in a global scale of gridding space 30o latitude and 2 km altitude found that 0.30, 0.98 and 0.96 in the lower stratosphere of tropics, mid and high latitudes respectively. Nevertheless, the correlation coefficient between MIPAS\_CH4\_224 and MIPAS\_N2O\_224 are 0.62, 0.80 and 0.66.

• Page 5 line 18: "The altitude range of the comparison [between MIPAS and FTIR] has been restricted to 18-21km". Does it make any sense to do that, given that the vertical resolution of a FTIR instrument is far worse than 3km?

Response: We agree that the sentence was not for the cases of comparison, it was the altitude ranges where the variability of water vapour and uncertainty of MIPAS CH4 had a relation. "The altitude range of the comparison has been restricted to 18-21 km." has been replaced by

"The altitude range of the comparison has been restricted to the lower altitude of MIPAS and upper altitude of FTIR sensitivity."

- Page 8 figure 2: This figure shows nicely how CH4 version 224 agrees better with FTIR than CH4 version 220. However, there are a number of fixes needed
  - The text on the figure may be a little small, especially on the colour bar. Text on a figure should end up about the same size as the caption text in the final version of the paper.
  - The colour bar does not match the colours in the dots. In particular, the dots include an orange colour which does not appear in the colour bar.
  - The cyan and green colours in the colour bar are not easy for the eye to distinguish. The yellow is very distinct from the other colours, but can be hard to see on a white background. Selecting a suitable colour scale is not trivial, and one should never be satisfied with the default colours in a plotting package.
  - The caption says that both upper and lower panels are version 220. It is clear from the text that the lower panels are version 224, so the caption needs correcting.
  - The altitudes in the colour bar are clearly in km; the caption should say so.
  - There is a linear fit line of some sort shown in the figure, but it is difficult to see and the caption does not say what it is for. Either the line should be made clearer and the caption should explain its purpose, or the line should be removed.
  - In plots such as this, where the two axes are for two estimates of the same quantity, and in the same units, the scale of the axes should be such that the 1:1 line is at 45 to the axes.
  - It is good practice to add labels (a), (b) etc. to panels of a multi-panel figure so that they can be referred to from the caption

Response on Page 8 figure 2: We have corrected all the points the referee had pointed out on figure 2 and its caption: Here is the new Fig. 2:

Figure 2. Scatter plot of daily mean values of MIPAS\_CH4\_220 (upper panel) and MIPAS\_CH4\_224 (lower panel) vs FTIR CH4 of Addis Ababa, Jungfraujoch and Ny-Ålesund sites from left to right respectively. The colour bar represents the altitude in Km. The thick red line is the best fit straight line while the black line would be obtained for a perfect agreement (CH4 (MIPAS) = CH4 (FTIR)). The correlation coefficients r of the MIPAS and FTIR series are summarized in table 1 (15-22 km).

To explore the coincident CH4 vmr measurements of MIPAS and FTIR, the scatter plots are shown in Fig. 2 at altitude range 15-30 km, color bar has been represented for 3 km space (blue, cyan, green, yellow and red). The high correlation at Jungfraujoch and Ny-Ålesund indicate that the instrument uncertainties are low relative to the methane variability for both MIPAS\_CH4\_220, MIPAS\_CH4\_224 with reduced the bias in the MIPAS\_CH4\_224 at Jungfraujoch. However, the correlation is less at Addis Ababa that indicates large instrument uncertainties relative to the methane variability. The correlation coefficients are shown in table 1 for altitude range 15-22 km, those are 0.84 (MIPAS\_CH4\_220) and 0.98 (MIPAS\_CH4\_224) for Jungfraujoch , 0.9 (MIPAS\_CH4\_220) and 0.89 (MIPAS\_CH4\_224) for Ny-Ålesund and less than 0.3 for both MIPAS\_CH4\_220, MIPAS\_CH4\_224 at Addis Ababa. Further, the data points of the new data version coincide better with the line with unity slope than the data of the older version. The latter tend to lie above this line for altitudes below 22 km for Addid Ababa and Jungfraujoch measurements. This confirms the findings of the previous section that low altitude methane in MIPAS\_CH4\_220 MIPAS data is biased high and this bias has been reduced in MIPAS\_CH4\_224.

The standard deviation of mean relative differences between MIPAS\_CH4\_220 and FTIR stated in the previous section also shows the precision and it is in the order of 4 to 8 % in altitude between 15 and 27 km of all the three sites and this is comparable to the estimated

precision reported by von Clarmann (2006). On the other hand, the precision between MIPAS\_CH4\_224 and FTIR is in the order of 1 to 5 %, maximum at Addis Ababa, tropics. Thus the effect of water vapour variability on the uncertainty of MIPAS  $CH_4$  (mainly on MIPAS\_CH4\_220) at tropics is required to asses in this work.

• Page 8, last two lines, running on to page 9: It seems very odd to do such a limited comparison of MIPAS and MLS H2O. Either the two instruments are known to agree well (in which case, why do a comparison at all?) or they are known to differ (in which case, how much can we learn from such a limited comparison?).

Response: Here, we use the comparison between MIPAS and MLS of  $H_2O$  so that to explore the variation of natural variability of water vapour over the three latitudinal bands. The only reason we had this comparison is to show that bias evaluation method is also important on investigating variability of  $H_2O$ . The overestimation of SD of mean relative difference at upper troposphere of the three sites is shown, but the overestimated value of SD of mean relative difference in tropics is much larger than the remaining two sites. This confirms the presence of large atmospheric variability of water vapour in tropics.

• Page 9 Lines 10–14: I am struggling to understand all this from the explanation given. We can see from figure 2 that CH4 has only a very small range of possible values at a given height in the tropical lower stratosphere. This will also be true of any chemically-stable tropospheric source gas, including N2O. The measurements could be unbiased and with relatively small random measurement error and you would still find them to have little correlation in a small altitude/latitude range. So the low correlations do not necessarily indicate large uncertainty, they may just indicate small true variability in both species. (Of course, both species decrease with height, so the correlation between them will appear good if you include data from a wide range of altitudes.)

Response: The description was indeed lacking some needed explanations in the main text. We added a new paragraph in section 4.3 that is:

Tracer-tracer correlation (i.e. CH4-N2O) measured by MIPAS has been used to show the variation of the uncertainty of MIPAS CH4 as a function of latitude and altitude. As revealed in different literatures (eg. Plumb et al., 2007; Payan et al., 2009), the correlation coefficients of long lived trace gases were also used to show the latitudinal and altitudinal variation of uncertainties of an instrument. Fig.4. Show the correlation coefficients between MIPAS CH4 220 and MIPAS\_N2O\_220, MIPAS\_CH4\_220 and MLS CH4. MIPAS\_CH4\_224 and MIPAS\_N2O\_224, and MIPAS\_CH4\_224 and MLS CH4. The correlation coefficients between MIPAS CH4 220 and MIPAS N2O 220 are only used to show the latitudinal variations of MIPAS\_CH4\_220 uncertainty. The larger the correlation of coefficient is the lesser the uncertainty and vice versa. We need only to show the uncertainty is different at different latitudinal bands. The correlation coefficient between MIPAS\_CH4\_220 and MIPAS\_N2O\_220 as a function of latitude and altitude are 0.30, 0.98 and 0.96 in the lower stratosphere over tropics, mid and high latitudes respectively.

Nevertheless, the correlation coefficient between MIPAS CH4 224 and MIPAS\_N2O\_224 are 0.62, 0.80 and 0.66. Hence, the result indicates the reduction of uncertainty of MIPAS\_CH4\_224 as its correlation in the lower stratosphere exceeds that of MIPAS\_CH4\_220.

• Page 9 lines 16–20: I do not understand what is meant by "subtracting the standard variability of water vapour from profiles of MIPAS CH4 220 and MLS methane" The authors need to explain what they did in more detail. A reader is supposed to be able (at least, in principle) to go away and repeat the work in a paper for himself. Based on this description, I would not be able to do so

Response on Page 9 lines 16–20: The contribution of water variability on the uncertainty of MIPAS  $CH_4$  has clearly seen in figure 4, we showed that the correlation between the two tracer gases derived from the instruments are expecting not to vary with latitude. However, the figure shows the correlation coefficient is less at lower stratosphere of tropics. On the other hand, this result confirms the large uncertainty of MIPAS\_CH4\_220 at lower stratosphere of tropics as obtained from bias evaluation and is due to the effects of atmospheric water variability.

The explanation written below has been placed in "4.3. Correlation plots of CH4 and N2O"

The influence of natural variability of water vapour on the uncertainty of MIPAS\_CH4\_220 and MIPAS\_CH4\_224 in the lower stratosphere of tropics with reduced effect on new version data. The contribution of water vapour variability to the large uncertainty of MIPAS\_CH4\_220 at lower stratosphere can be shown by the following assumption.

Assume that water vapour variability at the lower stratosphere of tropics has an effect on the amount of MIPAS CH4 profile. The large uncertainty of methane derived from MIPAS instruments in lower stratosphere of tropics is due to water vapour variability and this has shown in Fig. 5 by taking in to account the amount of water vapour variability that enhance the profile of methane in tropics using the equation below. Hence, the true concentration amount of MIPAS CH4 in the lower stratosphere is expressed as follows:

$$X_{t} = X_{m} - SD_{NV}$$
 5

Where  $X_t$  is the concentration amount after removing the effects of water vapour variability on the vmr amount of CH4 and N2O,  $X_m$  is the amount of methane obtained from the measurement and SDNV is the square root of estimated natural variability of H2O variance at upper troposphere and lower stratosphere (see Fig. 5).

The latitudinal variation of uncertainty of MIPAS\_CH4\_220 has been related with the variability of water vapour. However, the latitudinal variation of uncertainties of the new version data is reduced as water is jointly retrieved with methane. Fig. 5: shows the reduction of latitudinal variations of uncertainty after eq. 5 has been applied on the data sets. The correlation coefficient between MIPAS\_CH4\_220 and MIPAS\_N2O\_220 as a function of latitude and altitude after application of eq. 5, high variation of correlation coefficient has

been reduced as shown in Fig. 5. Thus indicates the effect of water vapour variability on the uncertainty of MIPAS\_CH4\_220 in lower stratosphere of tropics.

- Page 11 figure 4: If the authors decide to retain this figure, there are several improvements that could be made:
  - The title above each panel should be removed.
  - It is good practice to add labels (a), (b) etc. to panels of a multi-panel figure so that they can be referred to from the caption.
  - In general it is good practice to use a diverging (two-sided) colour scale such as that used here for a quantity which tends to be equally spaced about zero. But in this case there are no cells which have a negative correlation. The figure would therefore be clearer if a good single-sided scale were used, showing values only between zero and one. The caption could confirm that there are no negative correlations.

Response: We have corrected all the points the referee had pointed out on figure 4 after reorganizing the figures such as Fig. 4 only about exploring the variation of the uncertainty of MIPAS\_CH4\_220 and MIPAS\_CH4\_224 as a function of latitude and altitude and showing the reduction of uncertainty on MIPAS\_CH4\_224. Fig. 5 shows the uncertainty after removing the square root of water vapour variability variances from the amount of vmr values of  $CH_4$  and  $N_2O$  using eq. 5.

Figure 4: Correlation coefficients between (a) MIPAS\_CH4\_220 and MIPAS\_N2O\_220 (b) MIPAS\_CH4\_220 and MLS  $CH_4$  (c) MIPAS  $CH_4$  224 and MIPAS\_N2O\_224 (d) MIPAS\_CH4\_224 and MLS  $CH_4$  as a function of latitude and altitude for the period February 2010.

---

## Author Comment (AC2) · 19 Sep 2019

Responses to referee 3:   (received:  22 August 2019)

We thank the referee for very insightful questions and comments. They have helped to improve the quality of the paper. We have substantially revised and reorganized the manuscript, in many parts extended paragraphs has added. Our responses are given point-by-point below (blue Times New Roman font) following each of the reviewers' comments, which are repeated in full (black Times New Roman Italic font). Reproduced text from the revised manuscript is set in green Times New Roman font.

In addition to the change made in the manuscript to take into account your comments or the comments of the other referees, several other changes have been made and are listed here.

***General Comments***

*This manuscript presents an investigation of the large uncertainty of MIPAS V5R_CH4_220 methane retrievals in the tropical upper troposphere and lower stratosphere and attributes it to interference from atmospheric water vapour. Including water vapour when fit in methane reduces the uncertainties in the new data version MIPAS V5R_CH4_224.*

*The work is a useful contribution to the field and appropriate for AMT, but the manuscript needs major revisions and another round of reviews.*

*I have read the reviews of the other two referees. My assessment of the manuscript is similar to theirs, so I will not repeat their specific and technical comments.*

Response: The response to other referees has been responded and some of the response were also added here.

*In general, the manuscript should be improved in the following ways:*

Response: We would like to thank the reviewer for this positive evaluation. This paper has addressed the cause of large uncertainty in the old version of MIPAS CH4 in the upper troposphere and lower stratosphere of tropics which is due to water vapour variability. However, the contribution of water vapour in the uncertainty of new data sets of MIPAS CH4 had been reduced as water profile is jointly retrieved.

*Change the title. MIPAS is not a satellite. e.g., "The impact of H2O variability on the accuracy of MIPAS CH4 measurements [or retrievals] over the tropics:*

Response: As the referee suggest, we will replace the word "satellite" by "measurements" in the title and written as follows:   "Impacts of $H_2O$ variability on accuracy of MIPAS CH4 measurements over the tropics"

- *Rewrite the abstract for clarity, conciseness, and grammar. The explanation of the results is wordy and unclear.*

Response: In the first sentence of the abstract, before the period we added "at upper troposphere and lower stratosphere" to make clear where the uncertainties are large

Page1lines 4-6: has been replaced by "Coincident measurements by MIPAS, ground based FTIR and MLS of $CH_4$, $H_2O$ are used to estimate the standard uncertainty of MIPAS_CH4_220, MIPAS_CH4_224 and natural variability of $H_2O$. Moreover, MLS of $CH_4$ were derived from EOS MLS coincident measurements of atmospheric water vapour ($H_2O$), carbon monoxide (CO) and nitrous oxide ($N_2O$)."

Page1lines 6-8: has been replaced by " Different methods such as bias evaluation differential method and correlation analysis are employed to explore the latitudinal variations of standard uncertainty of MIPAS_CH4_220, MIPAS_CH4_224 and natural variability of water vapour."

Response: The results in the abstract have been rewritten as follows:

The averaged bias between MIPAS_CH4_220 and ground-based FTIR measurements in the altitude rang 15-22 km are 12.3%, 8.9 % and -1.2 % for tropics, mid-latitudes and high latitudes, respectively. Whereas the averaged bias for MIPAS_CH4_224 is 3.9 %, -2.8 % and -2.4 %. The average estimated uncertainties of MIPAS CH4 220 methane were obtained 5.9 %, 4.8 % and 4.7 % at altitude ranges of 15 to 27 km for tropics, mid-latitudes and high latitudes, respectively. On the other hand, the average estimated uncertainties of MIPAS CH4 224 methane were obtained 2.4 %, 1.4 % and 5.1 %. Moreover, the correlation coefficient between MIPAS CH4 220 and MIPAS V5R_N2O_220 in a global scale of gridding space 30 degree latitude and 3km altitude found that 0.30, 0.98 and 0.96 in the lower stratosphere of tropics, mid and high latitudes respectively. Nevertheless, the correlation coefficient between MIPAS CH4 224 and MIPAS V5R_N2O_224 are 0.62, 0.80 and 0.66.

*- The manuscript needs much clearer explanations of methods and results throughout, and more detail on how results were obtained.*

Response: As the referee suggested on clarity the manuscript in a way that readers can easy understand by adding detail explanation of methods and results.

P3L2-5: the sentences have been replaced by "The coincident measurements Of $H_2O$, $CH_4$ and $N_2O$ by MIPAS, ground based FTIR and MLS were used to estimate the uncertainty of MIPAS_CH4_220 and MIPAS_CH4_224 profiles and the natural variability of $H_2O$. MLS CH4 was derived from EOS MLS coincident measurements of atmospheric water vapour ($H_2O$), carbon monoxide (CO) and nitrous oxide ($N_2O$)."

Inserted after the period in P3L4; Different methods has applied to determine uncertainty of MIPAS_CH4_220, MIPAS_CH4_224 measurements and variability of water vapour at the three latitudinal bands. Intercomparison results of methane (CH4) measured by MIPAS with the ground based FTIR products obtained from Addis Ababa FTIR observatory and other two NDACC FTIR sites (Jungfraujoch, Switzerland and Ny-Ålesund, Spitsbergen). It has been analyzed using the statistical analysis methods detailed in von Clarmann (2006). Natural variability of water vapour and uncertainties of MIPAS methane can also be determined using differential method proposed by Fioletov et al. (2006) and applied on different literatures (Toohey et al. (2007); Sofieva et al. (2014) for the three atmospheric conditions

using at least two different measurement techniques. Furthermore, correlations analysis between.$CH_4$-$N_2O$ measured by MIPAS and MIPAS $CH_4$ and MLS $CH_4$ has been used to show the variation of the uncertainty of MIPAS $CH_4$ as a function of latitude and altitude in a global scale. Finally, the cause of high uncertainty of MIPAS_CH4_220 and its reduction in MIPAS_CH4_224 at the lower stratosphere of tropics has been assessed through taking its relation with water vapour variability using a regression analysis method.

Inserted after the period in P5L14; "Both the estimated standard deviation (SD) of instrument uncertainty (i.e. MIPAS $CH_4$ ) and standard deviation of water variability for a given location, time of year, and layer were obtained using equations 4. Applying equation (4) to these data sources creates two sets of SD of MIPAS $CH_4$ uncertainty estimates. Similarly, SD of water vapour variability was obtained for each of the three latitudinal bands. The value estimated SD uncertainty of MIPAS CH4 was calculated as square root of the mean variance estimates from the two data sources."

The following paragraph has been added as a last paragraph under methodology section so that to make clear the methods employed in the manuscript.

Replace the last paragraph in P5L15-20 by "In addition to the above methods employed in this paper, as the UT/LS, mixing ratios of these long-lived trace gases are largely controlled by dynamical processes, generally resulting in compact tracer-tracer correlations. These correlations are usually more compact in high and mid-latitudes, while in tropics a somewhat larger scatter is observed (Plumb et al., 2007; Payan et al., 2009). We used such methods to show the variation of MIPAS_CH4_220 uncertainty with high value at LS of tropics and its reduction in MIPAS_CH4_224 as a function of latitude and altitude in a global scale using corresponding values MIPAS_N2O_220, MIPAS_N2O_224 for February 2010. In addition, both version data sets of MIPAS $CH_4$ and MLS $CH_4$ version 3.3 for February 2010 have been discussed too. These correlations are calculated on latitude bins space by $30^O$ and on an altitude grid with 7 levels and spacing of 2 km."

*- Figures need to better describe what is shown and ALL plots and captions need revisions. e.g., Figure 1 shows (X - Y)/Z differences but doesn't say what X, Y, and Z are. Figures 4 needs a better colour scale, panel labels, y-axis label, larger fonts, etc. Figure 8 should plot CH4 vs. H2O, not H2O vs. CH4. Take a careful look at quality of all the figures.*

[Figure]

Response: The caption has been replaced by "Figure 1. Comparisons of MIPAS CH4 220 profile with FTIR (upper panel) and MIPAS CH4 224 profile with FTIR (lower panel). The relative differences (200*(FTIR VMR-MIPAS VMR)/(FTIR VMR + MIPAS VMR)) averaged over Addis Ababa, Jungfraujoch and NyÅlesund sites. Shaded area is the Standard deviation of the mean relative differences."

Response: We have corrected all the points the referee had pointed out on figure 4 after re-organizing the figures such as Fig. 4 only about exploring the variation of the uncertainty of MIPAS_CH4_220 and MIPAS_CH4_224 as a function of latitude and altitude and showing the reduction of uncertainty on MIPAS_CH4_224. Fig. 5 shows the uncertainty after removing the square root of water vapour variability variances from the amount of vmr values of $CH_4$ and $N_2O$ using eq. 5.

[Figure]

Figure 4: Correlation coefficients between (a) MIPAS_CH4_220 and MIPAS_N2O_220 (b) MIPAS_CH4_220 and MLS CH$_4$ (c) MIPAS CH4 224 and MIPAS_N2O_224 (d) MIPAS_CH4_224 and MLS CH$_4$ as a function of latitude and altitude for the period February 2010.

[Figure]

Figure 5: Correlation coefficients between (a) MIPAS_CH4_220 and MIPAS_N2O_220 (b) MIPAS_CH4_220 and MLS CH$_4$ (c) MIPAS CH4 224 and MIPAS_N2O_224 (d) MIPAS_CH4_224 and MLS CH$_4$ as a function of latitude and altitude for the period February 2010. After applying e.g.5 to remove the effect of water vapour variability on the latitudinal variation of the CC.

Figure 8. The random uncertainty of MIPAS CH4 220 (right) and MIPAS CH4 224 (left) versus the natural variability of $H_2O$ using a three years data sets, 2009-2011 for altitude 18-21 km of tropics.

*- Is it correct to extrapolate the results from three specific sites as representative of the tropics, mid-latitudes, and polar regions? Justify this assumption.*

Response: The three sites are found at different atmospheric conditions, different regions. Hence, doing an atmospheric research at Addis Ababa mean that, we are discussing and presenting results that represent tropics. The latitudinal bands where we consider in this paper represent the three regions while the FTIR data were used. Here in this paper, we even taking a global scale a analysis of latitudinal variation of MIPAS CH4 uncertainty and natural variability of water vapor (see Fig. 4 and Fig. 5).

*- Correlation seems be equated with cause, e.g., p16, para1.*

Response:  The detection of high uncertainty of MIPAS_CH4_220 in the lower stratosphere of tropics was related to water vapour variability as clearly seen in the correlation analysis which is Fig. 4 and Fig 5. Therefore, we have concluded that water vapour variability was the causal on the high uncertainty of MIPAS_CH4_220.

 *- Add a table stating sites/latitude bands used and time periods.*

Response: The tables in page 8 and page 9, which are results summery of the bias evaluation and scatter plot presented in this work. As you suggested, we improved the tables by adding a column that explains about latitudinal bands of the location, time period and number of coincidence are stated.

*- Use consistent terminology when referring to the MIPAS versions, e.g., both V5R_CH4_220 and MIPAS CH4 220 are used. Use the former throughout, and similarly for V5R_CH4_224.*

Response: corrected

*- Section 4.3 should be rewritten for clarity.*

Response: The description was indeed lacking detail interpretation of the method, some needed explanations in the main text. We added a new paragraph that can make it clear to understand the important of the section.

Tracer-tracer correlation (i.e. CH4-N2O) measured by MIPAS has been used to show the variation of the uncertainty of MIPAS CH4 as a function of latitude and altitude. As revealed in different literatures (eg. Plumb et al., 2007; Payan et al., 2009), the correlation coefficients of long lived trace gases were also used to show the latitudinal and altitudinal variation of uncertainty of instrument. Figure 4. Show the correlation coefficients between MIPAS_CH4_220 and MIPAS_N2O_220, MIPAS_CH4_220 and MLS $CH_4$, MIPAS_CH4_224 and MIPAS_N2O_224, and MIPAS_CH4_224 and MLS $CH_4$. The correlation coefficients between MIPAS_CH4_220 and MIPAS_N2O_220 are only used to

show the latitudinal variations of MIPAS CH4 220 uncertainty. The larger the correlation of coefficient is the lesser the uncertainty and vice versa. We need only to show the uncertainty is different at different latitudinal bands. The correlation coefficient between MIPAS_CH4_220 and MIPAS_N2O_220 as a function of latitude and altitude are 0.30, 0.98 and 0.96 in the lower stratosphere over tropics, mid and high latitudes respectively. Nevertheless, the correlation coefficient between MIPAS_CH4_224 and MIPAS_N2O_224 are 0.62, 0.80 and 0.66. Hence, the result indicates the reduction of uncertainty of MIPAS_CH4_224 as its correlation in the lower stratosphere exceeds that of MIPAS_CH4_220.

The explanation written below has been placed in "4.3. Correlation plots of $CH_4$ and $N_2O$"

The influence of natural variability of water vapour on the uncertainty of MIPAS_CH4_220 and MIPAS_CH4_224 in the lower stratosphere of tropics with reduced effect on new version data. The contribution of water vapour variability to the large uncertainty of MIPAS_CH4_220 at lower stratosphere can be shown by the following assumption.

Assume that water vapour variability at the lower stratosphere of tropics has an effect on the amount of MIPAS CH4 profile. The large uncertainty of methane derived from MIPAS instruments in lower stratosphere of tropics is due to water vapour variability and this has shown in Fig. 5 by taking in to account the amount of water vapour variability that enhance the profile of methane in tropics using the equation below. Hence, the true concentration amount of MIPAS $CH_4$ in the lower stratosphere is expressed as follows:

$$X_t = X_m - SD_{NV} \hspace{3cm} 5$$

Where $X_t$ is the concentration amount after removing the effects of water vapour variability on the vmr amount of $CH_4$ and $N_2O$, $X_m$ is the amount of methane obtained from the measurement and $SD_{NV}$ is the square root of estimated natural variability of $H_2O$ variance at upper troposphere and lower stratosphere (see Fig. 5).

The latitudinal variation of uncertainty of MIPAS_CH4_220 has been related with the variability of water vapour. However, the latitudinal variation of uncertainties of the new version data is reduced as water is jointly retrieved with methane. Fig. 5: shows the reduction of latitudinal variations of uncertainty after eq. 5 has been applied on the data sets. The correlation coefficient between MIPAS_CH4_220 and MIPAS_N2O_220 as a function of latitude and altitude after application of eq. 5, high variation of correlation coefficient has been reduced as shown in Fig. 5. Thus indicates the effect of water vapour variability on the uncertainty of MIPAS_CH4_220 in lower stratosphere of tropics.

*- Data providers who are co-authors don't need to be thanked in the Acknowledgements.*

Response: The acknowledgement has changed as follows:

We greatly acknowledge the MLS science teams for the satellite data used in this study. Special thanks go to Dr. samuel takele for his contribution on calibrating the spectra

measured by the FTIR at Addis Ababa. Finally, authors would like to thank Mekelle and Addis Ababa universities for the sponsorship and financial supports.

*- The References should be revised to ensure that they are correct and have consistent formatting. Some have incomplete information and several are old AMTD references (e.g., Laeng et al., 2015; Sepulveda et al., 2012). AMTD references are to manuscripts under review. These should be updated to the published AMT references.*

Response: The references have been corrected as follows:

Errera, Q., Ceccherini, S., Christophe, Y., Chabrillat, S., Hegglin, M. I., Lambert, A., Ménard, R., Raspollini, P., Skachko, S., van Weele, M., 15 and Walker, K.A.: Harmonization and Diagnostics of MIPAS ESA CH4 and N2O Profiles Using Data Assimilation; Atmos. Meas. Tech. Discuss., doi:10.5194/amt-2016-245, 2016.

Errera, Q., Ceccherini, S., Christophe, Y., Chabrillat, S., Hegglin, M. I., Lambert, A., Ménard, R., Raspollini, P., Skachko, S., van Weele, M., and Walker, K.A.: Harmonization and Diagnostics of MIPAS ESA CH4 and N2O Profiles Using Data Assimilation; Atmos. Meas. Tech.,9,5895-5909, doi:10.5194/amt-9-5895-2016, 2016

Laeng, A., Plieninger, J., von Clarmann, T., Stiller, G. , Eckert, E., Glatthor, N., Grabowski, Haenel, N., Kiefer, M., Kellmann, S., Linden, A., Lossow, S., Deaver, L., Engel, A., Harvig, M., Levin, I., McHugh, M., Noel, G., and Walker, K.: Validation of MIPAS IMK/IAA methane profiles, Atmos. Meas. Tech. Discuss., 8, 5565–5590, doi:10.5194/amtd-8-5565 2015.

Laeng, A., Plieninger, J., von Clarmann, T., Stiller, G. , Eckert, E., Glatthor, N., Grabowski, Haenel, N., Kiefer, M., Kellmann, S., Linden, A., Lossow, S., Deaver, L., Engel, A., Harvig, M., Levin, I., McHugh, M., Noel, G., and Walker, K.: Validation of MIPAS IMK/IAA methaneprofiles, Atmos. Meas. Tech., 8, 5251–5261, doi:10.5194/amt-8-5251-2015, 2015.

Sepûlveda, E., Schneider, M., Hase, F., Garcîa, O., E., Gomez-Pelaez, A., Dohe, S., Blumenstock, T., and Guerra, J., C.,: Long-term validation 10 of total and tropospheric column-averaged CH4 mole fractions obtained by mid-infrared ground-based FTIR spectrometry, Atmos. Meas. Tech. Discuss., 5, 1381–1430, 2012.

Sepûlveda, E., Schneider, M., Hase, F., Garcîa, O., E., Gomez-Pelaez, A., Dohe, S., Blumenstock, T., andGuerra, J., C.,: Long-term validation of total and tropospheric column-averaged CH4 mole fractions obtained by mid-infrared ground-based FTIR spectrometry, Atmos. Meas. Tech., 5, 1425–1441, doi:10.5194/amt-5-1425-2012, 2012

*- The manuscript needs line-by-line copy editing to correct the many typographical, grammatical, and technical errors.*

Response: Done

*- Overall, the manuscript is poorly written. It needs a complete rewrite for scientific clarity. I encourage all of the authors to review the next version carefully prior to resub-mission*

Response: As you suggested, we have added and some re-arrangements of the manuscript by adding an extended sentences and paragraphs on the manuscript so that it attains scientific clarity. We have put it below the changes made on the manuscript and those that is not stated under the comments of the referee.

---

## Author Comment (AC3) · 19 Sep 2019

Responses to referee 2:  (received:  12 July 2019)

We thank the referee for very insightful questions and comments. They have helped to improve the quality of the paper. We have substantially revised the manuscript, in many parts extended paragraphs has added. Our responses are given point-by-point below (blue Times New Roman font) following each of the reviewers' comments, which are repeated in full (black Times New Roman Italic font). Reproduced text from the revised manuscript is set in green Times New Roman font.

In addition to the change made in the manuscript to take into account your comments or the comments of the other referees, several other changes have been made and are listed here.

Paper review

*This paper shows that water vapor variability impacted the methane concentration retrievals from MIPAS 220 and is improved in the 224 version. The paper uses a methodology of using coincident measurements from two data sources to derive the atmospheric variability and each instrument's random noise contribution following a technique published by Fioletov 2006. First here are some major issues I have with the paper. Given the basic assumptions the Fioletov method I believe is OK however I think the authors need to consider some possible limitations. One being that the instrument noise may depend on the concentration amount of methane due to forward model nonlinearities. This can be checked by correlating the retrieved amount against its reported uncertainty supplied by the MIPAS team. Secondly when deriving the instrument noise estimate from the coincident data, it would be interesting to compare that to the value supplied with the data set as a validation of the method. Neither of those were not done.*

Response:  This paper has addressed the causes of large uncertainty of MIPAS_CH4_220 in the upper troposphere and lower stratosphere of tropics which is due to water vapour variability. However, the contribution of water vapour in the uncertainty of new data sets of MIPAS_CH4_224 had been reduced as water profile is jointly retrieved.

We added a sentence:  The noise that is a random error of the instrument didn't have any relation to the amount of methane concentration on the upper troposphere and lower stratosphere.

*The presentation is not clear in many places and needs to be reworked a lot in order to publish this. Figure 4 for example shows a before and after like correlation analysis for methane version 220. In the after figure the authors say they subtracted water vapor variability from the CH4 retrieval and show how much better the correlation has improved. I have no idea how you can after the fact remove the water vapor variabilty impacts from the 220 data set or the 224 (figure 5) data set (which itself is significantly improved in this regard due to the simultaneous retrieval of H2O and CH4). Because the authors provide no explanation of how this is done, I am recommending rejection. The "removing" the effect of H2O interference leading to greatly improved agreement with correlative data is the central*

*point of the paper and establishment of cause described in the title, this needs to be much better explained*

Response on Page 9 lines 16–20: The contribution of water variability on the uncertainty of MIPAS $CH_4$ has clearly seen in figure 4, we showed that the correlation between the two tracer gases derived from the instruments are expecting not to vary with latitude. However, the figure shows the correlation coefficient is less at lower stratosphere of tropics. On the other hand, this result confirms the large uncertainty of MIPAS_CH4_220 at lower stratosphere of tropics as obtained from bias evaluation and is due to the effects of atmospheric water variability.

The explanation written below has been placed in "4.3. Correlation plots of $CH_4$ and $N_2O$"

The influence of natural variability of water vapour on the uncertainty of MIPAS_CH4_220 and MIPAS_CH4_224 in the lower stratosphere of tropics with reduced effect on new version data. The contribution of water vapour variability to the large uncertainty of MIPAS_CH4_220 at lower stratosphere can be shown by the following assumption.

Assume that water vapour variability at the lower stratosphere of tropics has an effect on the amount of MIPAS CH4 profile. The large uncertainty of methane derived from MIPAS instruments in lower stratosphere of tropics is due to water vapour variability and this has shown in Fig. 5 by taking in to account the amount of water vapour variability that enhance the profile of methane in tropics using the equation below. Hence, the true concentration amount of MIPAS $CH_4$ in the lower stratosphere is expressed as follows:

$$X_t = X_m - SD_{NV} \qquad\qquad 5$$

Where $X_t$ is the concentration amount after removing the effects of water vapour variability on the vmr amount of $CH_4$ and $N_2O$, $X_m$ is the amount of methane obtained from the measurement and $SD_{NV}$ is the square root of estimated natural variability of $H_2O$ variance at upper troposphere and lower stratosphere (see Fig. 5).

The latitudinal variation of uncertainty of MIPAS_CH4_220 has been related with the variability of water vapour. However, the latitudinal variation of uncertainties of the new version data is reduced as water is jointly retrieved with methane. Fig. 5: shows the reduction of latitudinal variations of uncertainty after eq. 5 has been applied on the data sets. The correlation coefficient between MIPAS_CH4_220 and MIPAS_N2O_220 as a function of latitude and altitude after application of eq. 5, high variation of correlation coefficient has been reduced as shown in Fig. 5. Thus indicates the effect of water vapour variability on the uncertainty of MIPAS_CH4_220 in lower stratosphere of tropics.

*The text was hard to follow and comprehend and I give a few examples of this below.*

*On page 9 line 18 there is a reference to a middle panel in figure 4 a figure with 4 panels in a 2X2 arrangement. What is the middle panel?*

Response: The error has been created while we taking the first figure prepared in the draft with three panel.. Now, it has been corrected.

*Figure 7 show profiles of H2O variability at three station sites. The profile at some altitudes is clipped at zero suggesting that it either is negative (not possible) or unknown. One profile it is exactly zero which is extremely unlikely (ie absolutely no atmospheric variability).*

Response on Page 12 lines 6-8 and page 13 figure 7: Fig. 7 shows the natural variability of H2O determined using Eq. 4 from the data MIPAS V5R_H2O_220 and MLS (Ver3.3) for the time period of February 2010. The resulting natural variability at 15-17 km altitude are 8.4 %, 5.4 % and 3.4 % for Addis Ababa,Jungfraujoch, and NyAlesund. Here in the figure the variability of water vapour becomes negative that shows an over estimation of random errors of the instruments are exist as the natural variability is a combination of the three estimated sample variances and had large uncertainty  (see Eq. 4). Moreover, natural variability of water vapour could also find from the MIPAS instrument using

*I don't know why the figure 8 scatter plot of monthly averages in one height range for the 224 data set contains many more the 36 points for 3 year monthly averages*

Response:  In our previous manuscript, we used averaged on each level such as 18, 19, 20 and 21 km. now corrected as averaged values between 18-21km

The following sentence has been added in page 14 L10:  The number of points is 36  in both scatter plot that represent the monthly averaged SD natural variability of water vapour and uncertainty of MIPAS_CH4_220 and MIPAS_CH4_224 in the altitude ranges of 18-21 km. for time period of three year.

[Figure]

Figure 8: The scatter plot between standard deviation of natural variability of $H_2O$ and the random uncertainty of MIPAS_CH4_220 (right) and MIPAS_CH4_224 (left) using a three years data sets, 2009-2011 for altitude 18-21 km and latitudinal band of tropics in the lower stratosphere.

*A sentence on page 13, line 15 seems to refer to a figure not included in the paper. It cannot possibly be describing figure 8.*

Page 13 lines 15: While we prepared this manuscript, we were trying to put the relation between the uncertainty of MIPAS $CH_4$ and variability of water vapour using a correlation coefficient between SD uncertainty of MIPAS $CH_4$ and SD of water vapour variability as function of altitude. Finally, we decided to put it in scatter plot at the lower stratosphere of tropics.

*There are a lot wording and grammer errors in here that need improving.*

Response: we have taking in to consideration all the referee has suggested for improving the manuscript, such major corrections has been posted below.

Response: The results in the abstract have been rewritten as follows:

The averaged bias between MIPAS_CH4_220 and ground-based FTIR measurements in the altitude rang 15-22 km are 12.3%, 8.9 % and -1.2 % for tropics, mid-latitudes and high latitudes, respectively. Whereas the averaged bias for MIPAS_CH4_224 is 3.9 %, -2.8 % and -2.4 %. The average estimated uncertainties of MIPAS CH4 220 methane were obtained 5.9 %, 4.8 % and 4.7 % at altitude ranges of 15 to 27 km for tropics, mid-latitudes and high latitudes, respectively. On the other hand, the average estimated uncertainties of MIPAS CH4 224 methane were obtained 2.4 %, 1.4 % and 5.1 %. Moreover, the correlation coefficient between MIPAS CH4 220 and MIPAS V5R_N2O_220 in a global scale of gridding space 30 degree latitude and 3km altitude found that 0.30, 0.98 and 0.96 in the lower stratosphere of tropics, mid and high latitudes respectively. Nevertheless, the correlation coefficient between MIPAS CH4 224 and MIPAS V5R_N2O_224 are 0.62, 0.80 and 0.66.

P3L2-5: the sentences have been replaced by "The coincident measurements Of $H_2O$, $CH_4$ and $N_2O$ by MIPAS, ground based FTIR and MLS were used to estimate the uncertainty of

MIPAS_CH4_220 and MIPAS_CH4_224 profiles and the natural variability of H$_2$O. MLS CH4 was derived from EOS MLS coincident measurements of atmospheric water vapour (H$_2$O), carbon monoxide (CO) and nitrous oxide (N$_2$O)."

Inserted after the period in P3L4; Different methods has applied to determine uncertainty of MIPAS_CH4_220, MIPAS_CH4_224 measurements and variability of water vapour at the three latitudinal bands. Intercomparison results of methane (CH4) measured by MIPAS with the ground based FTIR products obtained from Addis Ababa FTIR observatory and other two NDACC FTIR sites (Jungfraujoch, Switzerland and Ny-Ålesund, Spitsbergen). It has been analyzed using the statistical analysis methods detailed in von Clarmann (2006). Natural variability of water vapour and uncertainties of MIPAS methane can also be determined using differential method proposed by Fioletov et al. (2006) and applied on different literatures (Toohey et al. (2007); Sofieva et al. (2014) for the three atmospheric conditions using at least two different measurement techniques. Furthermore, correlations analysis between.CH$_4$-N$_2$O measured by MIPAS and MIPAS CH$_4$ and MLS CH$_4$ has been used to show the variation of the uncertainty of MIPAS CH$_4$ as a function of latitude and altitude in a global scale. Finally, the cause of high uncertainty of MIPAS_CH4_220 and its reduction in MIPAS_CH4_224 at the lower stratosphere of tropics has been assessed through taking its relation with water vapour variability using a regression analysis method.

Inserted after the period in P5L14; "Both the estimated standard deviation (SD) of instrument uncertainty (i.e. MIPAS CH$_4$ ) and standard deviation of water variability for a given location, time of year, and layer were obtained using equations 4. Applying equation (4) to these data sources creates two sets of SD of MIPAS CH$_4$ uncertainty estimates. Similarly, SD of water vapour variability was obtained for each of the three latitudinal bands. The value estimated SD uncertainty of MIPAS CH4 was calculated as square root of the mean variance estimates from the two data sources."

The following paragraph has been added as a last paragraph under methodology section so that to make clear the methods employed in the manuscript.

Replace the last paragraph in P5L15-20 by "In addition to the above methods employed in this paper, as the UT/LS, mixing ratios of these long-lived trace gases are largely controlled by dynamical processes, generally resulting in compact tracer-tracer correlations. These correlations are usually more compact in high and mid-latitudes, while in tropics a somewhat larger scatter is observed (Plumb et al., 2007; Payan et al., 2009). We used such methods to show the variation of MIPAS_CH4_220 uncertainty with high value at LS of tropics and its reduction in MIPAS_CH4_224 as a function of latitude and altitude in a global scale using corresponding values MIPAS_N2O_220, MIPAS_N2O_224 for February 2010. In addition, both version data sets of MIPAS CH$_4$ and MLS CH$_4$ version 3.3 for February 2010 have been discussed too. These correlations are calculated on latitude bins space by 30$^O$ and on an altitude grid with 7 levels and spacing of 2 km."